# Nested order-disorder framework containing a crystalline matrix with self-filled amorphous-like innards

Kejun Bu [1], Qingyang Hu [1], Xiaohuan Qi[2], Dong Wang[1], Songhao Guo[1], Hui Luo[1], Tianquan Lin[2], Xiaofeng Guo [3], Qiaoshi Zeng [1], Yang Ding [1], Fuqiang Huang [2], Wenge Yang [1], Ho-Kwang Mao [1] & Xujie Lü [1] ✉

Solids can be generally categorized by their structures into crystalline and amorphous states with different interactions among atoms dictating their properties. Crystalline-amorphous hybrid structures, combining the advantages of both ordered and disordered components, present a promising opportunity to design materials with emergent collective properties. Hybridization of crystalline and amorphous structures at the sublattice level with long-range periodicity has been rarely observed. Here, we report a nested order-disorder framework (NOF) constructed by a crystalline matrix with self-filled amorphous-like innards that is obtained by using pressure to regulate the bonding hierarchy of $Cu_{12}Sb_4S_{13}$. Combined in situ experimental and computational methods demonstrate the formation of disordered Cu sublattice which is embedded in the retained crystalline Cu framework. Such a NOF structure gives a low thermal conductivity (~0.24 $W \cdot m^{-1} \cdot K^{-1}$) and a metallic electrical conductivity ($8 \times 10^{-6}$ $\Omega \cdot m$), realizing the collaborative improvement of two competing physical properties. These findings demonstrate a category of solid-state materials to link the crystalline and amorphous forms in the sublattice-scale, which will exhibit extraordinary properties.

Properties of materials are determined by the interaction among atoms and can be tuned through the structural flexibility of different building components. Depending on the structural arrangements, solid-state materials are classified into crystalline with long-range periodicity and amorphous with only short- to medium-range ordering[1–4]. Integrating both states by creating crystalline-amorphous hybrid materials has been a long-standing research interest. These hybrid materials could possess advantageous properties from both crystalline and disordered units, which are increasingly attractive for potential technological applications, including black $TiO_2$ nanomaterials for photocatalysis[5,6], two-dimensional electron gases at crystalline-amorphous oxide interfaces for transparent conductors[7], metal-organic frameworks (MOFs) and their composites for catalysis[8]. From the local structure point of view, hybridization has been made at the mesoscopic scale[9–11], such as paracrystalline silicon[12], intermediate crystalline metallic glass[13], and melted chains in high-pressure metals[14–16]. However, structural design at the sublattice level of crystalline-amorphous hybrid materials with long-range periodicity is still challenging and, as far as we know, has been rarely realized. Regulating the chemical-bond hierarchy of crystals enables the design of hybrid structures based on periodic sublattices and thus provides an opportunity for the discovery of material forms with emergent or enhanced properties[17,18].

Due to variable coordination conditions and valence states, copper chalcogenides have large structural variability and exhibit an intrinsic chemical-bond hierarchy, which gives a high and anisotropic tunability[18–20]. Besides chemical tailoring, the degree of bonding

[1]Center for High Pressure Science and Technology Advanced Research, Shanghai 201203, China. [2]CAS Key Laboratory of Materials for Energy Conversion, Shanghai Institute of Ceramics, Chinese Academy of Sciences, Shanghai 200050, China. [3]Department of Chemistry and Alexandra Navrotsky Institute for Experimental Thermodynamics, Washington State University, Pullman, WA 99164, USA. ✉e-mail: xujie.lu@hpstar.ac.cn

hierarchy can be tuned by applying external stimuli, including temperature, pressure, and electric field[17,18]. Recently, temperature-induced hybrid state has been reported in $Cu_2Se$ where the $Cu^+$ sublattice becomes amorphous on warming and induced liquid-like flow[19,21]. Besides, the amorphous-to-crystal transition can be triggered by electric pulses in phase-change memory material $Ge_2Sb_2Te_5$ with bonding energy hierarchy[17,22]. However, strong vibration of all atoms at high temperature or electric field leads to the whole structural instability and second-phase precipitation, which limits the tunability and formation of crystalline-amorphous hybrid structures[23]. As a state variable, pressure provides an effective and clean approach to adjust the atomic interactions and thus alter the bonding configuration without changing chemical compositions[24–27]. Therefore, pressure processing enables the exploration and modulation of crystalline-amorphous hybrid structures which would collaboratively optimize the competing physical properties.

In this work, to realize the designed structure by tuning crystal sublattices, we select a nested copper chalcogenide $Cu_{12}Sb_4S_{13}$ which possesses a strong chemical-bond hierarchy[28]. The structure contains a rigid framework and a set of weakly bonded atoms serving as the soft sublattice, endowing it with anisotropic tunability of the sublattices under external stimuli. By using in situ high-pressure diagnostics, we systematically investigate the variations of bonding configuration, lattice structure, thermal and electrical properties. Pressure processing creates a disordered sublattice embedded in the retained crystalline matrix, forming a nested order-disorder framework (NOF). Such a NOF structure exhibits the theoretical minimum lattice thermal conductivity and metallic electrical conductivity. We further quantitatively describe the variation of sublattice from crystalline to NOF and its effects on the physical properties with the help of first-principles calculations.

## Results
### Pressure-induced formation of NOF structure
Tetrahedrite $Cu_{12}Sb_4S_{13}$ possesses a cubic sphalerite-like structure of $I\text{-}43m$ symmetry with 6 of the 12 Cu atoms occupying tetrahedral $12d$

sites (Cu1) at the vertices of equal-edge truncated octahedrons (Archimedean solid) and the remaining six Cu2 atoms distributed on trigonal planar $12e$ sites (Fig. 1). Each Sb atom is bonded to three S atoms, giving a space in the structures occupied by the lone pair electrons (LPEs). The structure features two structural motifs, which consist of the $[Cu_{12}S_{24}]$ rigid framework (Cu1 framework) and the rattling $Sb[Cu2S_3]Sb$ soft sublattice (Fig. 1a). The bonding hierarchy exists in the $Sb[Cu2S_3]Sb$ unit, where Sb atoms stay away from the Cu2 atoms and form weak interactions compared with Cu2−S covalent bonds. Such a bonding hierarchy induces quasi-localized and large amplitude anharmonic Cu2 vibrations akin to swing-like rattling modes, resulting in a dynamically flexible sublattice with emergent properties[28–30]. By regulating this bonding asymmetry, pressure could effectively adjust the sublattices and would create hybrid materials.

In situ single-crystal and powder X-ray diffraction (XRD) measurements were performed to investigate the structural evolution of $Cu_{12}Sb_4S_{13}$ under pressure. The Rietveld analysis results for representative XRD data are shown in Supplementary Table 1 and Fig. 1. Upon compression, all Bragg diffraction spots and peaks shift to higher two-theta angles due to the lattice contraction (Fig. 2a, b). When the applied pressure exceeds 12 GPa, the diffraction spots broaden obviously and the diffuse diffraction halos appear, implying the onset of partial disordering. At 16.5 GPa, most diffraction spots disappear with several single-crystal XRD spots on the amorphous scattering pattern (Fig. 2a), indicating the formation of the crystalline-amorphous hybrid state. The powder XRD results show a few Bragg diffraction peaks on the broad diffuse background beyond 16 GPa (Fig. 2b), which further support the pressure-induced hybridization of crystalline and amorphous structures. It is worth noting that the retained Bragg diffraction spots and peaks are associated with the structure of Cu1 framework (Figs. 1a and 2b). In other words, the rigid Cu1 framework retains crystalline above 16 GPa, while the rattling $Sb[Cu2S_3]Sb$ interior subunit becomes disordered, resulting in the NOF structure with an ordered matrix and the self-filled amorphous-like innards (as illustrated in Fig. 1b).

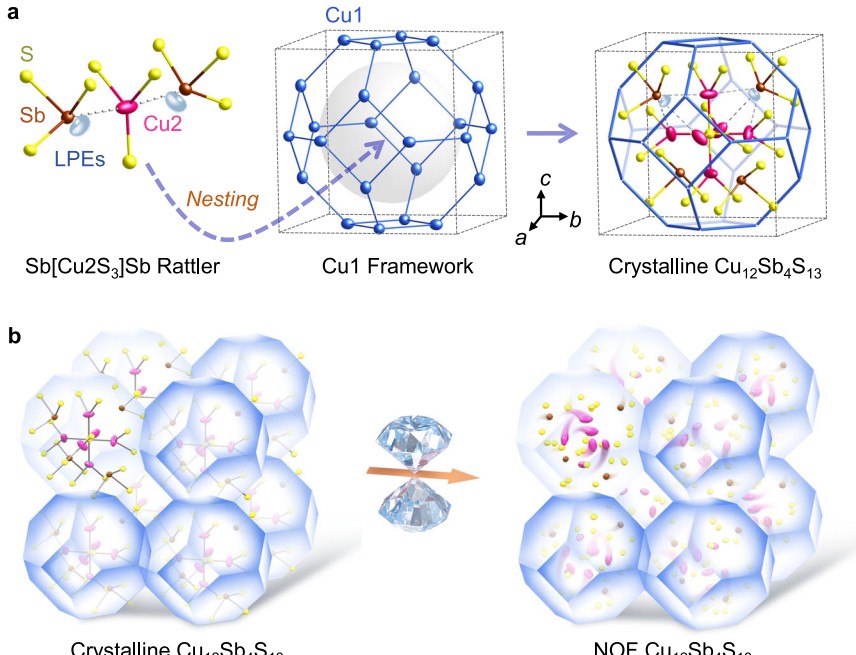

**Fig. 1 | Illustration of the $Cu_{12}Sb_4S_{13}$ structure with bonding asymmetry and the formation of nested order-disorder framework (NOF). a** $Cu_{12}Sb_4S_{13}$ is composed of the $[Cu_{12}S_{24}]$ rigid framework and the nested rattling $Sb[Cu2S_3]Sb$ unit. The lobe-like balls represent lone pair electrons (LPEs). The S atoms in Cu1 framework are omitted for clarity. **b** Pressure-induced transformation of crystalline $Cu_{12}Sb_4S_{13}$ to the NOF structure. The increasing anharmonic Cu vibration during compression causes a disordered Cu2 sublattice embedded in the retained Cu1 crystalline matrix, creating the NOF.

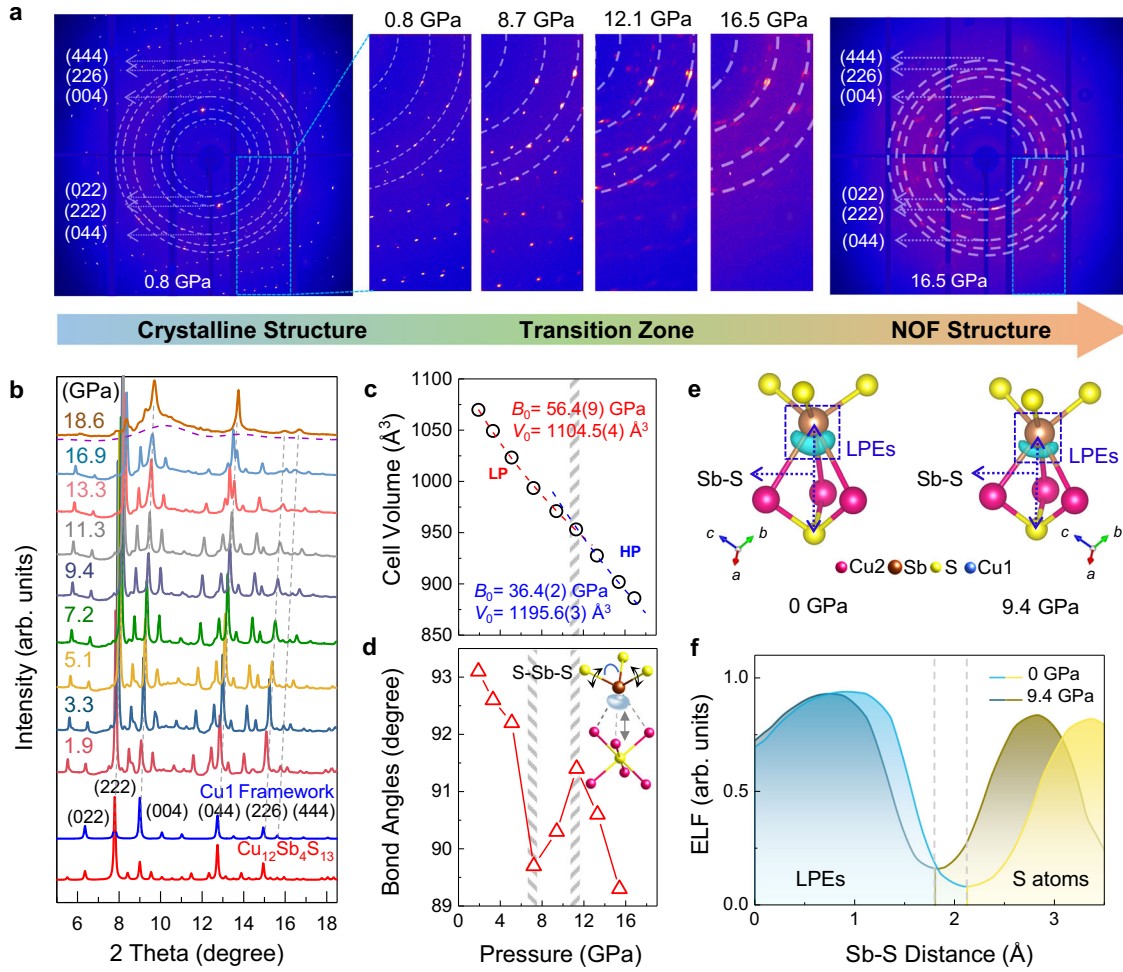

**Fig. 2 | Structural characterization of Cu₁₂Sb₄S₁₃ with in situ synchrotron diffractions and electron localization function (ELF) calculations. a** The single-crystal XRD images and (**b**) powder XRD patterns of Cu₁₂Sb₄S₁₃ at selected pressures. The diffraction spots from the single-crystal diamond are masked. The retained Bragg diffraction spots and peaks beyond 16 GPa indicate that the Cu1 framework keeps crystalline under high pressure. **c** Unit-cell volume during compression. The values of bulk modulus $B_0$ were determined to be 56.4(9) and 36.4(2) GPa in the low-pressure (LP) and high-pressure (HP) regions, respectively. **d** Bond angles of S−Sb−S as a function of pressure. The inset shows the strong nonlinear repulsive force from the suppressed lone pair electrons (LPEs) on Cu2 atoms under high pressure. **e** The ELF graphs and (**f**) the ELF line profiles between Sb and S atoms at ambient pressure and at 9.4 GPa.

The pressure-induced variation of unit-cell volume is shown in Fig. 2c, which reveals unusual compressibility. By fitting the Birch–Murnaghan equation of state (Eq. 1 and Supplementary Eq. 1), the values of bulk modulus $B_0$ were determined to be 56.4(9) and 36.4(2) GPa in the low- and high-pressure regions, respectively. Such an abnormal decrease of $B_0$ indicates the more compressible structure under high pressures, which is related to the large atom displacement parameter (ADP) of Cu2[28]. As shown in Supplementary Fig. 2, the ADP values of Cu2 considerably increase from 0.020 to 0.064 Å² during compression, indicating the enhanced rattling vibration and the moveable Cu2 atoms under high pressures. In the low-pressure region, both the strong Cu1 framework and the relatively weak ordered Cu2 sublattice support the structure against compression and contribute to high bulk modulus. Under high pressure, the movable Cu2 atoms induce the collapse of Sb[Cu2S₃]Sb sublattice and no longer provide help, resulting in the anomalous compressibility. The stability of Sb[Cu2S₃]Sb sublattice with bonding hierarchy is sensitive to pressure-induced changes in the behavior of LPEs. As shown in Fig. 2d, a sudden change of S−Sb−S bond angles occurs at 7 GPa, which can be ascribed to the suppression of LPEs during compression[31]. In situ Raman spectroscopy also shows an increase in slope of Raman shift for Sb−S bending mode (*E*) beyond 7 GPa (Supplementary Fig. 3), confirming the suppression of LPEs that changes S−Sb−S bond angle (see the detailed

discussion in the Supplementary Discussion). To further elucidate the nature of the interaction between LPEs and Cu2 sublattice, we investigated the variation of chemical bonding by employing electron localization function (ELF) and Bader charge[32,33]. The isosurface of a lobe shape charge extension around the Sb atoms indicates the existence of LPEs. During compression, the LPEs are suppressed over 7 GPa (Fig. 2e, f and Supplementary Fig. 4), which is consistent with the XRD results. To quantify the bond order between atom pairs, density derived electrostatic and chemical charge (DDEC) based on Bader charges were employed[34,35]. As shown in Supplementary Table 3, the bond order between Cu2 and out-of-plane Sb atoms is 0.10 at 0 GPa and significantly increases to 0.22 at 9.4 GPa, which confirms the enhanced electrostatic force during compression. Such an enhanced electrostatic repulsive force from LPEs shoves Cu2 atoms away from the equilibrium position, which converts the rattling Cu2 atoms to be diffusing and thus gives rise to the disordered Cu2 sublattice.

To investigate the local structure of NOF structure, we examined the atomic-scale structures of pristine and high-pressure treated samples using spherical aberration-corrected scanning transmission electron microscopy (STEM). Figure 3a and Supplementary Fig. 5 show the high-angle annular dark-filed (HAADF) STEM images of Cu₁₂Sb₄S₁₃ projected along the [1 1 −2] and [0 2 −1] zone-axis at ambient condition, with neighboring atom columns of Sb/Cu and Cu1/Cu2. After the high-

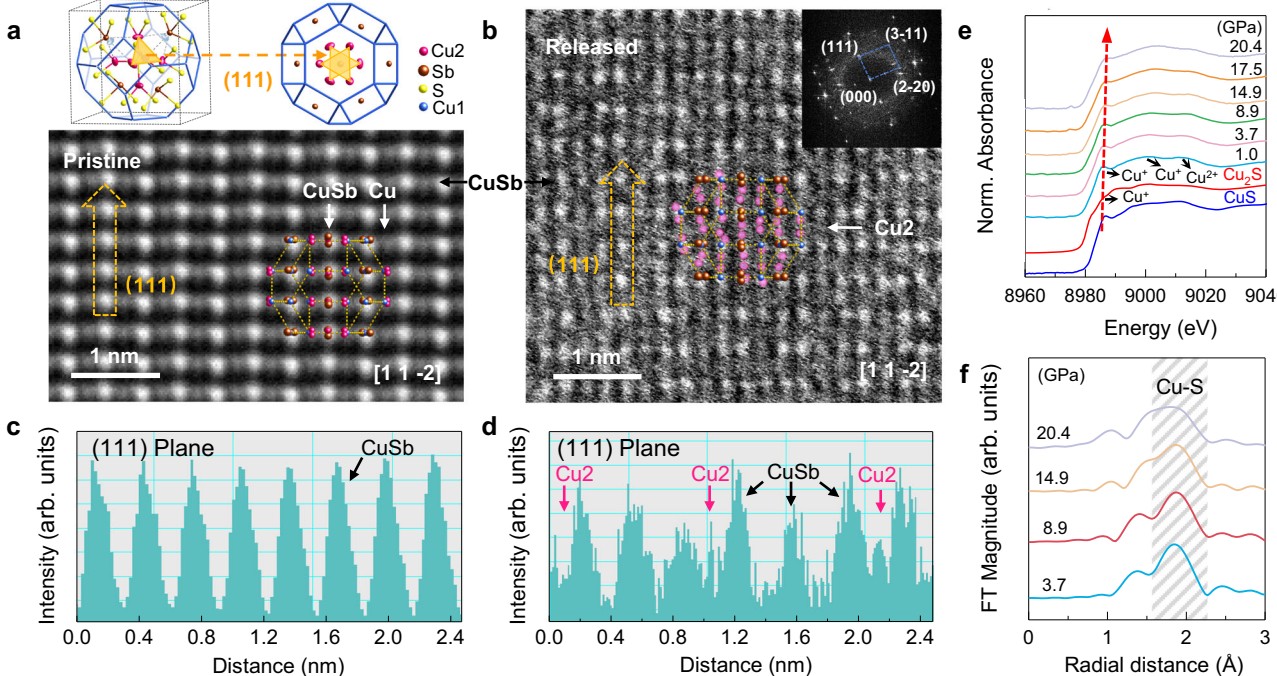

**Fig. 3 | Atomic-scale analysis and atomic coordination environment of Cu₁₂Sb₄S₁₃.** The HAADF STEM images taken along the [1 1 −2] zone-axis of (**a**) the initial sample at ambient condition and (**b**) the sample treated by high pressure. The top panel of (**a**) shows the (1 1 1) planes that are related to the Cu2 atoms. The inset of (**b**) shows the corresponding select area electron diffraction. The retained electrons diffraction spots with the broad diffuse background confirm the hybrid structure. Intensity-scan profiles taken from the (1 1 1) plane with Sb/Cu atom columns of (**c**) the initial sample at ambient and (**d**) the sample treated by high pressure, as indicated by a yellow arrow in (**a**) and (**b**). **e** The near edge region of XAS on the K-edge of Cu for Cu₁₂Sb₄S₁₃ at different pressures. **f** Fourier transform (FT) curves of the EXAFS data of Cu₁₂Sb₄S₁₃.

pressure treatment, the disordering occurs within the (1 1 1) planes, which is related to the Cu2 atoms (Fig. 3b). From the intensity-scan profiles, one can see a good periodicity for the initial sample (Fig. 3c). While after high-pressure treatment, the background lifts and additional peaks can be observed which corresponding to the randomly occupied Cu2 atoms (Fig. 3d). This observation indicates that Cu2 atoms move away from the equilibrium position and become disordered in the crystalline matrix. Moreover, as shown in the select area electron diffraction (inset of Fig. 3b), the retained diffraction spots corresponding to the structure of Cu1 framework sit on the broad diffuse background, which is consistent with the XRD results in Fig. 2, confirming the formation of NOF structure.

Furthermore, in situ high-pressure X-ray absorption spectra (XAS) were collected to understand the evolution of atomic coordination environment (Fig. 3e and Supplementary Fig. 6a). In the analysis of the near edge region of XAS, the Cu ions exhibit mixed-valences of Cu⁺ and Cu²⁺ in Cu₁₂Sb₄S₁₃ (Fig. 3e). The obviously weakened peak at around 8983 eV beyond 8.9 GPa implies the weakening of the Cu2−S bonding. Figure 3f and Supplementary Fig. 6b show the Fourier transform (FT) plots and k-weight of the extended X-ray absorption fine structure (EXAFS), respectively. The distances around 1.5−2.3 Å are considered as common Cu−S covalent bonds (Fig. 3f)[36,37]. The obviously weakened and broadened Cu−S peaks beyond 8.9 GPa indicate the widely distributed Cu−S bond lengths and complex coordination environment of Cu2 atoms, which confirms the distorted innards in NOF structure.

### Low lattice thermal conductivity of the NOF Cu₁₂Sb₄S₁₃

Such a crystalline-amorphous hybrid structure, having a periodically ordered matrix with the disordered filling, could possess extraordinary physical properties like thermal and electrical conductivity. The total thermal conductivity is derived from two contributions: the carrier thermal conductivity ($\kappa_e$) and the lattice thermal conductivity ($\kappa_L$)[38]. The $\kappa_e$ is determined by the variation of electrical resistivity $\rho$ using the

Wiedemann−Franz law with the ideal Lorenz number (Supplementary Eq. 5)[38]. As shown in Fig. 4a, the $\rho$ decreases during compression and then increases from 7 to 13 GPa which is caused by the LPEs-induced high vibration of Cu2 sublattice (details in the Supplementary Discussion). The enhanced vibration significantly increases lattice scattering and suppresses electron transport. The corresponding $\kappa_e$ has a negligible contribution on the total thermal conductivity beyond 11 GPa ($\kappa_e < 20\%$ $\kappa_{total}$)[38], suggesting that $\kappa_L$ dominates the thermal conductivity when the Cu2 sublattice becomes disordered (Fig. 4b and Supplementary Fig. 7).

To elucidate the role of the pressure-induced disordered sublattice on thermal property, $\kappa_L$ values were determined by fitting the full widths at half maximum of the Raman optical mode (Supplementary Eqs. 3 and 4, Supplementary Figs. 8 and 9)[39]. The obvious decrease of $\kappa_L$ after 7 GPa is attributed to the suppression of LPEs (Fig. 4b), which enhances rattling-like vibrations of Cu2 sublattice via repulsive force. Owing to the highly dynamic fluctuations of Cu2 atoms caused by the lone-pair effects, $\kappa_L$ drops to a very low value over 11 GPa (0.24 W·m⁻¹·K⁻¹ at 12.6 GPa). Such a low value reaches the theoretical minimum lattice thermal conductivity of Cu₁₂Sb₄S₁₃, which is generally only found in glass-like solids where the phonon mean free path approximates to the interatomic spacing[40]. Furthermore, we determined the pressure dependence of total thermal conductivity ($\kappa_R$) based on the in situ high-pressure Raman scattering measurements (Supplementary Fig. 7)[39]. As shown in Supplementary Figs. 11 and 12, the Sb−S bending mode (E) of Cu₁₂Sb₄S₁₃ exhibits relatively strong dependences on temperature and laser power, which can be used to calculate $\kappa_R$ (Eq. 2 and Supplementary Eq. 2). $\kappa_R$ increases during compression up to 7 GPa and decreases thereafter (Fig. 4c), whose variation is in line with the thermal conductivity calculated by $\kappa_e + \kappa_L$. Consequently, the pressure-regulated bonding hierarchy creates the NOF structure with amorphous-like Cu2 innards that yields low thermal conductivity.

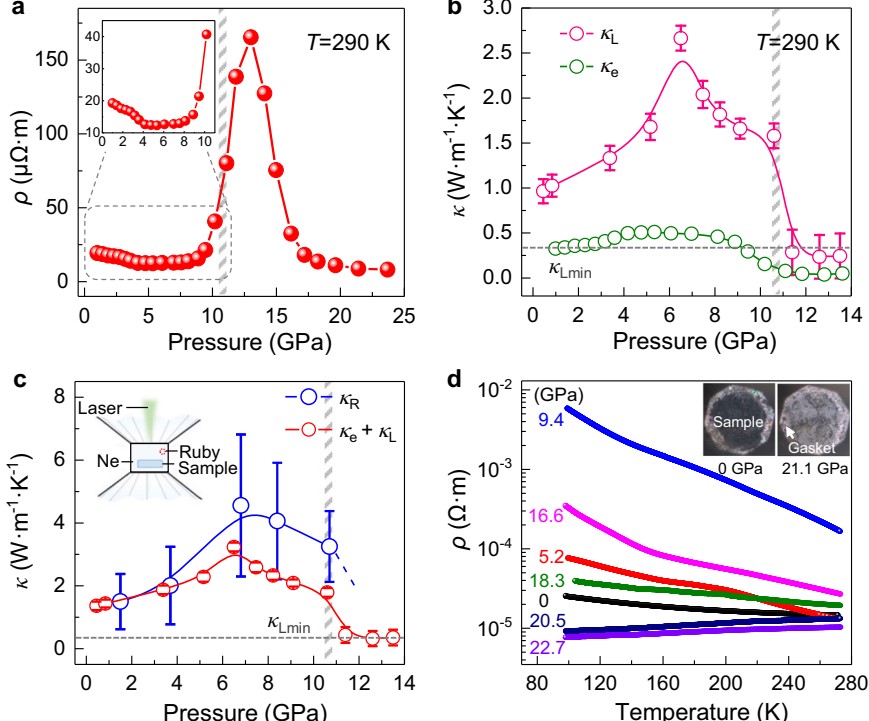

**Fig. 4 | The electrical and thermal properties of Cu₁₂Sb₄S₁₃ at different pressures. a** Pressure dependence of the electrical resistivity ($\rho$) at 290 K. The inset shows a zoomed $\rho$ in the lower pressure region. **b** Pressure dependence of the lattice thermal conductivity ($\kappa_L$) and the electronic thermal conductivity ($\kappa_e$). The dash line is a theoretical minimum lattice thermal conductivity ($\kappa_{Lmin}$) of Cu₁₂Sb₄S₁₃. **c** The values of thermal conductivity at various pressures were determined by two different methods. The error bars are the uncertainties of the first-order temperature (laser power) derivatives of the phonon frequencies. The inset shows the diagram for the thermal conductivity measurements using in situ high-pressure Raman scattering method. The details of experimental methods are given in the Supplementary Methods. **d** Temperature-dependent electrical resistivity ($\rho$-T) at different pressures. The semiconductor-to-metal transition occurs during compression. The inset shows the optical images in reflection mode at 0 and 21.1 GPa.

## Pressure-induced metallization

To further explore the electronic properties of the NOF structure, we investigated the optical and electrical properties at different pressures. Cu₁₂Sb₄S₁₃ is a narrow bandgap semiconductor with an indirect energy gap of 1.23 eV at ambient pressure (Supplementary Fig. 13)[41]. The optical absorption edge becomes unobservable at 9.8 GPa, which is associated with the disordering of Cu2 sublattice and the potential pressure-induced metallization (details in the Supplementary Discussion). To demonstrate the pressure-induced metallization, we examined the temperature-dependent resistivity ($\rho$-T) at various pressures and a semiconductor-to-metal transition is observed. As shown in Fig. 4d, a positive d$\rho$/dT at high pressure indicates the metallic feature of Cu₁₂Sb₄S₁₃ ($8 \times 10^{-6}$ $\Omega$·m at 22.7 GPa), where the crystalline Cu1 matrix in NOF serves as mixed-valent electron conducting channel that often leads to the metallic behavior[28,42,43]. Moreover, Cu₁₂Sb₄S₁₃ shows increased reflectance in near-infrared range at high pressures (Supplementary Fig. 14), which indicates a rising carrier concentration[44,45]. In addition, the sample became more reflective at high pressure which is similar to the color of the T301 steel gasket (inset of Fig. 4d). Therefore, a metallic state of the NOF Cu₁₂Sb₄S₁₃ is realized at room temperature. Taking together with the low thermal conductivity, the two competing transport properties (thermal and electrical) for thermoelectrics can be concurrently optimized in such a NOF material.

## Discussion

By analyzing the structural and physical characteristics of Cu₁₂Sb₄S₁₃, we have demonstrated the achievement of NOF structure with low thermal conductivity and metallic electrical conductivity by regulating the bonding hierarchy. We have stated the effects of LPE variations on the Cu2 sublattice disordering. With the knowledge of chemical

bonding, we further performed first-principles molecular dynamic (FPMD) simulations to examine the dynamic behaviors of atoms under pressure[46]. The vibrational density of states (VDOS) of all atoms at 0 and 9.4 GPa are shown in Supplementary Fig. 17. The low-lying modes (<100 cm⁻¹) are mainly attributed to Cu2 atoms, which signify weak bonding duo to their low frequencies[28]. The other atoms contribute to the higher-energy modes and shift towards higher energy with pressure increasing, which suggests more rigid bond formation under pressure[28]. Whereas, the low-lying modes corresponding to Cu2 atoms still stay at the low-frequency region, which retain weak bonding features under high pressures. During compression, the bonding hierarchy induces the destruction of the sublattice with Cu2 weak bonds but the rest crystalline framework retains, resulting in the formation of NOF structure.

The trajectory of Cu atoms in MD simulations in Cu₁₂Sb₄S₁₃ under different pressures is shown in Fig. 5a and Supplementary Fig. 18. The flexible [Cu2S₃] coordination brings more anharmonic motion of Cu2 atoms than Cu1 atoms and such movement of Cu2 is greatly promoted under high pressure. To better visualize the anharmonicity, the mean square displacement (MSD) in the time domain was obtained from the FPMD trajectory[47]. With equilibrium established, the MSD values of Cu2 approach a higher plateau than those of other atoms (Fig. 5b), suggesting that Cu2 sublattice is the major contribution to disordered state under pressure. The increasing MSD values of Cu2 during compression imply more anharmonic and disordered sublattice (Fig. 5c and Supplementary Fig. 19). According to the MSD parameter, the melting point of the sublattice can be roughly determined by the Lindemann criterion (Eq. 5)[48]. The Lindemann melting parameters ($\delta$) of Cu2 atoms were 0.13, 0.21, and 0.25 at 0, 9.4, and 13.3 GPa, respectively. The high-pressure $\delta$ values exceed the

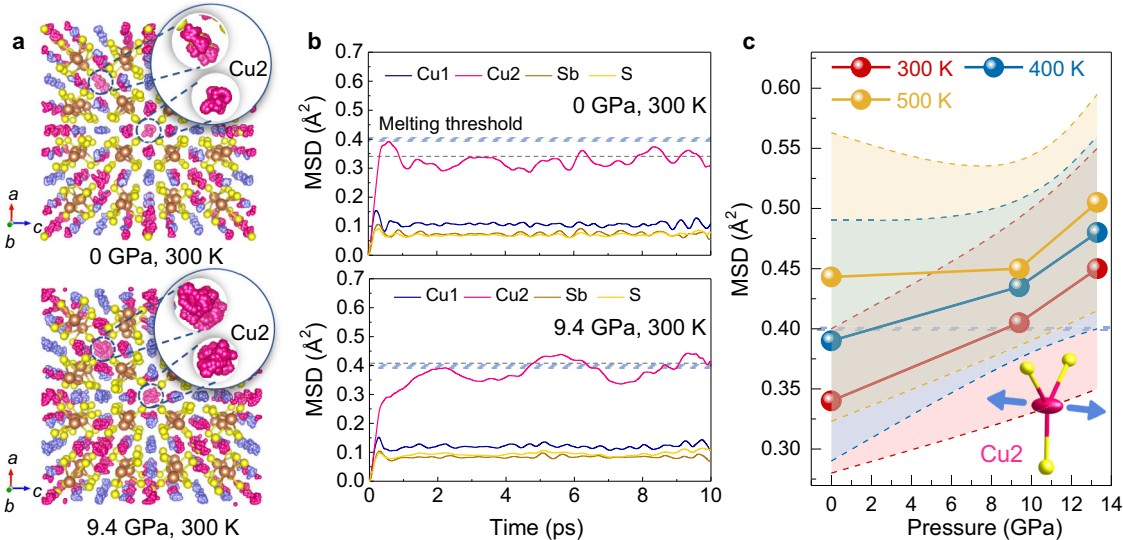

**Fig. 5 | First-principles molecular dynamic (FPMD) simulations of $Cu_{12}Sb_4S_{13}$ at different pressures. a** The trajectory of Cu atoms and (**b**) time-dependent mean square displacement (MSD) at 0 GPa and 9.4 GPa at room temperature. The dash and shade lines are the average MSD and melting threshold MSD of Cu2 atoms, respectively. **c** The MSD of Cu2 atoms in $Cu_{12}Sb_4S_{13}$ at different pressures and temperatures.

maximum melting threshold value of 0.2, indicating the formation of glass-like Cu2 sublattice. Noted that the MSD values of Cu2 atoms at higher temperatures are less sensitive with pressure increasing (Fig. 5c, Supplementary Figs. 18 and 20). Although both the temperature and pressure tend to stimulate vibration and motion of all atoms, the pressure more selectively regulates the local bonding to induce disordered Cu2 sublattice that creates the NOF structure (Fig. 5c and Supplementary Fig. 21). Such a crystalline-amorphous hybrid exhibits properties including low thermal conductivity and metallic electrical conductivity in $Cu_{12}Sb_4S_{13}$. Therefore, pressure has created a form of solid with combined features of amorphous and crystalline states by regulating the bonding configuration.

In summary, our combined experimental-computational results have demonstrated a matter state of nested order-disorder framework (NOF) that combines the crystalline and amorphous features in periodic sublattice. Using pressure to regulate the bonding hierarchy, the obtained NOF structure of $Cu_{12}Sb_4S_{13}$ contains amorphous-like Cu2 innards embedded in the crystalline Cu1 matrix. Such a NOF structure simultaneously achieves a low lattice thermal conductivity (0.24 $W\cdot m^{-1}\cdot K^{-1}$) and a metallic electrical conductivity ($8 \times 10^{-6}$ $\Omega\cdot m$). This work sheds light on the design of crystalline-amorphous hybrid materials with emergent collective properties.

## Methods
### Sample preparation
Single crystals of $Cu_{12}Sb_4S_{13}$ were synthesized by a solvent-thermal method with ethylene glycol (EG) acting as the solvent. 6 mmol of $CuCl_2\cdot 3H_2O$ (AR, 99%), 2 mmol of $SbCl_3$ (AR, 99%), 45 mmol of Thiourea (AR, 99.9%) powder, and 50.0 mL of EG were added into a 100 mL beaker to form a solution under stirring. The reaction mixture was then transferred into a 100 mL Teflon-lined stainless autoclave and heated at 200 °C for seven days. Finally, a great number of black tetrahedron-shaped crystals were obtained after cooling down to room temperature.

### In situ high-pressure characterizations
The high-pressure characterizations of $Cu_{12}Sb_4S_{13}$ were provided by symmetrical diamond anvil cells (DACs). Type II-a diamonds were chosen to measure the UV-Vis absorption spectroscopy and Raman spectroscopy, while type I-a diamonds were used to perform the X-ray

diffractions (XRD) and resistance measurements. The pressure was determined by the ruby fluorescence method[49].

### Synchrotron XRD measurements
The in situ single-crystal XRD at high pressures was carried out at the experimental station 13 BM-C (GSECARS) of Advanced Photon Source (APS), Argonne National Laboratory (ANL). Silicon oil was used as pressure transmitting medium in the single-crystal XRD experiments[50]. The wavelength of the monochromated X-ray beam was 0.434 Å (28.6 keV). In situ powder XRD experiments were performed at beamline 16 ID-B of HPCAT at APS, ANL. Neon was used as the pressure transmitting medium in the powder XRD experiments. The wavelength of the monochromatic X-ray beam is 0.4066 Å (30.5 keV). The sample-to-detector distance and other parameters of the detector were calibrated using the $CeO_2$ standard. The diffraction images were integrated using the Dioptas program and structure refinements were carried out by using the Rietveld method in FullProf software[51,52]. The cell volume data are fitted by the Birch-Murnaghan equation of state:

$$P(V) = \frac{3B_0}{2}\left[\left(\frac{V_0}{V}\right)^{\frac{7}{3}} - \left(\frac{V_0}{V}\right)^{\frac{5}{3}}\right]\left\{1 + \frac{3}{4}(B'-4)\left[\left(\frac{V_0}{V}\right)^{\frac{2}{3}} - 1\right]\right\} \quad (1)$$

where $V_0$ is the initial volume at ambient pressure, $V$ is the compressed volume, $B_0$ is the bulk modulus, and $B'$ is the derivative of the bulk modulus with respect to pressure[53].

### UV−Vis−NIR absorption spectroscopy
The UV-Vis-NIR absorption spectra of $Cu_{12}Sb_4S_{13}$ ranging from 600 nm to 1500 nm were collected by using a Xeon light source and the home-designed spectroscopy system (Gora-UVN-FL, built by Ideaoptics). Silicone oil was used as the pressure transmitting medium.

### Raman spectroscopy
Raman spectra were collected using a Renishaw Raman microscope. A laser excitation wavelength of 532 nm was utilized. A single crystal $Cu_{12}Sb_4S_{13}$ was chosen to measure, and spectra were collected in the range of 100–600 $cm^{-1}$. Before measurement, the silicon wafer at 520 $cm^{-1}$ was chosen to calibrate the Raman system. Neon was used as the pressure transmitting medium.

## Transmission electron microscopy

The pristine and high-pressure treated samples were examined. The scanning transmission electron microscopy (STEM) specimens were prepared by focused ion beam (FIB, Helios nanolab 600, FEI, USA). The atomic-scale high-angle annular dark-field (HAADF) STEM images were carried out on a spherical aberration-corrected Hitachi HF5000 operating at 200 kV.

## X-ray absorption spectra measurement

In situ high-pressure X-ray absorption spectra (XAS) of Cu K-edge were collected at BL05U station in Shanghai Synchrotron Radiation Facility (SSRF). The energy dispersive mode was used for studies of materials under high pressure. To avoid DACs glitches, polycrystalline diamond anvils were used for XAS measurements under pressure. The XAS data of the samples were collected at different pressure in transmission mode. Internal energy calibration was accomplished by measuring the standard Cu foil. The acquired XAS data were processed according to the standard procedures using the Athena implemented in the IFEFFIT software packages[54].

## Resistance measurement

The resistance values of $Cu_{12}Sb_4S_{13}$ single crystal were determined using a Keithley 6221 current source and an 2182 A nanovoltmeter. NaCl powders were used as the pressure-transmitting medium. The temperature variation of the resistance was measured by using the liquid nitrogen cooling system and collected on Keithley meters.

## Thermal conductivity measurement

Power-dependent and temperature-dependent Raman scattering measurements were used to measure pressure dependence of thermal conductivity ($\kappa$) as previously reported[55]. The $\kappa$ is calculated by the following formula:

$$\kappa = \frac{2\alpha}{\pi r_0} \frac{\chi_T}{\chi_W} \qquad (2)$$

where $\alpha$ and $r_0$ are the absorptance and the width of laser beam on the single crystal, respectively. The $\chi_T$ and $\chi_W$ are the first-order temperature and laser power derivative of phonon frequency shift, respectively. Neon was used as the pressure transmitting medium to avoid the heat dissipation[56].

## Electronic structure calculations

Density Functional Theory (DFT) calculations were performed using the Vienna Ab Initio Simulation Package (VASP). Perdew–Burke–Ernzehof (PBE) exchange-correlation function of the generalized gradient approximation (GGA) was chosen for the exchange and correlation terms. We used a Γ-centered k-points grid of $8 \times 8 \times 8$ and the plane wave with 450 eV cutoff energy to relax crystal structures. The electron localization function (ELF) is described by the following equation[32,33]:

$$\mathrm{ELF}(r) = \left\{ 1 + \left[ \frac{K(r)}{Kh(\rho(r))} \right]^2 \right\}^{-1} \qquad (3)$$

where $\rho(r)$ is the electron density, $K$ is the curvature of the electron pair density and $Kh[\rho(r)]$ corresponds to a uniform electron gas with spin-density equal to the local value of $\rho(r)$. Bader charges were calculated to give a uniquely defined way to partition the electron density around each atom. The density-derived electrostatic and chemical charge (DDEC) methods based on Bader charges are employed to calculate the bond orders[34,35].

## First-principles molecular dynamics simulation

The same pseudopotential and PBE functional were employed in the FPMD simulation. Here, a single gamma-point (Γ) was adopted for k-points sampling molecular dynamics. We doubled the x-axis of the lattice to construct a $2 \times 1 \times 1$ supercell, which contains a total number of 116 atoms. The simulation ran under a constant number of atoms, volume, and temperature (NVT) ensemble, as well as a constant number of atoms, pressure, and temperature (NPT) ensemble. Along the trajectory, we now initialize simulation with 3 ps NVT simulation for heating (near 0 K to 300–500 K), with 1 fs for each step and temperature controlled by a Nosé–Hoover thermostat[47], then follow by 5 ps NPT to the target pressure and eventually run 10 ps NVT simulation for equilibrium. The standard deviations of pressure are generally less than 1 GPa. Reaching equilibrium generally takes $10^4$ FPMD steps (10 ps), which is judged by the fluctuation of thermodynamical variables. After achieving equilibrium, the system continued to run $10^4$ FPMD steps. The results were extracted to evaluate the motion of Cu, Sb, and S atoms. The mean square displacement (MSD) is averaged over atoms of each type:

$$\langle [\vec{r}(t)]^2 \rangle = \frac{1}{N} \sum_{i=1}^{N} \langle [\vec{r_i}(t+t_0) - \vec{r_i}(t_0)]^2 \rangle \qquad (4)$$

where $\vec{r_i}(t)$ is the displacement of the $i$ atom at time $t$, and N is the total number of atoms in the system. The Lindemann parameter can be defined as[48]:

$$\delta = \frac{\sqrt{\langle \Delta r^2 \rangle}}{a} \qquad (5)$$

The $\Delta r^2$ is the MSD parameter of atoms and $a$ is the nearest-neighbor distance.

## Data availability

The X-ray crystallographic structures reported in this study have been deposited at the Cambridge Crystallographic Data Centre (CCDC), under deposition number 2189334-2189342. These data can be obtained free of charge from The Cambridge Crystallographic Data Centre via www.ccdc.cam.ac.uk/data_request/cif. The data supporting the key findings of this study are available within the article and its Supplementary Information files. Any further relevant data are available from the corresponding authors on request.

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

## Acknowledgements

This work is supported by the National Nature Science Foundation of China (NSFC) (Grant Nos. U1530402 and 51527801). Portions of this

work were performed at Sector 13 (GeoSoilEnviroCARS, The University of Chicago) and Sector 16 (HPCAT), Advanced Photon Source (APS), Argonne National Laboratory. GeoSoilEnviroCARS is supported by the National Science Foundation-Earth Sciences (EAR-1634415) and Department of Energy-GeoSciences (DE-FG02-94ER14466), and partially by COMPRES under NSF Cooperative Agreement EAR-1606856. This research used resources of the Advanced Photon Source, a U.S. Department of Energy (DOE) Office of Science User Facility operated for the DOE Office of Science by Argonne National Laboratory under Contract No. DE-AC02-06CH11357. The high-pressure XAS measurements were performed at Sector BL05U of the Shanghai Synchrotron Radiation Facility (SSRF). Q.H. is supported by the CAEP Research Project (CX20210048) and a Tencent Xplorer prize. The authors thank Drs. Xiaojia Chen and Yan Zhou for their helpful suggestion and discussion on the in situ measurements of thermal conductivity. The authors thank Chenxi Zhu for his technique assistance with STEM measurements.

## Author contributions

K.B. and X.L. conceived the project. K.B., X.Q. and T.L. synthesized single-crystal samples. X.G. and X.L. collected the high-pressure X-ray diffraction data. K.B. and Q.H. carried out the first-principles calculations. K.B., H.L. and S.G. analyzed the high-pressure thermal and electrical transport data with the assistance of D.W.; K.B. collected the high-pressure XAS data with the assistance of W.Y.; K.B. analyzed the STEM data with the help of X.Q. and X.L.; K.B. and X.L. wrote the manuscript and revised by F.H., W.Y., Y.D., Q.Z. and H.-K.M. All authors have interpreted the findings, commented on the paper, and approved the final version.

## Competing interests

The authors declare no competing interests.
