## [Peer Review File · Nature Communications]

Nested order-disorder framework: A crystalline matrix with self-filled amorphous-like innardsREVIEWER COMMENTS

Reviewer #1 (Remarks to the Author):

Bu *et al.* report the formation of a nested order-disorder framework in $\text{Cu}_{12}\text{Sb}_4\text{S}_{13}$ compound, induced by an application of high pressure. The system was thoroughly studied by a range of experimental and theoretical methods including synchrotron X-ray diffraction, Raman and UV-Vis-NIR spectroscopy, first-principles molecular dynamics simulations, *etc.*

The authors claim that they have discovered a new category of solid-state materials, constructed of a crystalline matrix with self-filled amorphous innards. I disagree with this statement. First, the formation of amorphous guest in a crystalline host matrix is a well-known phenomenon in high-pressure metals (see, for example, McMahon *et al.* 10.1103/PhysRevLett.93.055501, McBride *et al.* 10.1103/PhysRevB.91.144111). Although, ‘melted’ chains in high-pressure metals are not nested, there is a clear similarity of this phenomenon with the disorder of $\text{Cu}_{12}\text{Sb}_4\text{S}_{13}$ substructure. Unfortunately, the Authors do not even mention this in their manuscript. Metal-organic or metal-inorganic frameworks with strongly disordered guest molecules are also not uncommon in scientific literature.

Apart from that there are several issues with the data presentation, which I would like to point out:

1. On Page 4 the Authors write that the structural softening “is related to the large atom displacement parameter of Cu_2 ”. In the experimental section the Authors write that Rietveld refinement was performed (Figure S1). Therefore, it should be possible to track the development of the refined Cu_2 ADP with pressure increase. This information should be given in the text.
2. Following the previous comment, if Rietveld refinement was performed at all pressure points, the corresponding *cif* files must be provided in the Supplementary Information.

3. How was the disorder modelled in the Rietveld refinements at high pressures (Figure S1, 13.3 and 16.9 GPa)?

Reviewer #2 (Remarks to the Author):

In this manuscript, Bu et al. report the design of a nested order/disorder copper chalcogenide ($\text{Cu}_{12}\text{Sb}_4\text{S}_{13}$) structure via the application of external high pressure. The concept of the study is very interesting and the objective of the quest of a hybrid crystalline/amorphous material, which combines the features of both states simultaneously in a structure, is timely and of significant importance.

The authors have used a range of experimental techniques, which are described in details in the Supplementary Information. In addition, density-functional theory (DFT) simulations have been used to provide some theoretical evidence. Moreover, the authors performed some extensive analyses, and from different perspectives, to examine their experimental and simulation data in order to verify their view. Therefore, it is important to highlight and appreciate their efforts as reported in this study.

My main concern based on the results is how valid is the claim of the realization of this hybrid amorphous/crystalline material. A structure with increased disorder does not necessarily mean that it is a glassy structure. For example, the cubic (rocksalt) crystalline structure of the chalcogenide phase-change memory material, $\text{Ge}_2\text{Sb}_2\text{Te}_5$, is a highly-disordered crystalline structure. Such a structure, does not have any resemblance with an amorphous, glass-like structure. Another example are the grain boundaries, where two different crystalline structures, in a polycrystalline material, form a specific defective interface.

Throughout the manuscript there are also some certain things that require further clarification:

-> I think the Introduction is poor. There is only one paragraph where the authors are trying to build up the objective. I would prefer, as a reader, to get more convinced about the importance of the design of such hybrid structures. For instance, there are no applications mentioned of these materials. How one can utilize their enhanced properties in specific applications? Why it is important to tailor and exploit the features of both states regarding the operation of devices? Tackling these questions should strengthen the arguments of the study.

-> The authors use the terminology of a void, which is a specific term in amorphous materials to define certain space. Also, certain rules (e.g. Voronoi polyhedra, and others) are typically employed to calculate and characterize voids. The term is used rather loosely in the manuscript and it needs either justification or revision.

-> Pressure was utilized to create the hybrid structure. Such choice makes sense and I do not have any objection. In modern literature has been demonstrated that the application of an external electric field can modify the atomic and electronic structures of crystalline and amorphous materials. Have the authors considered such an option? It should be discussed as means of modifying structures and tailoring properties, at least.

-> The NOF structure reported in Figure 1b is it from real data (trajectory/positions) or it is a sketch/graphic? It is not clear. If it corresponds to the simulation trajectory, from the MD simulations, then the argument is more valid. In contrast, if it is a sketch I am afraid it is not enough. It is also rather blurred.

-> The authors mention in the text softening of the structure without really measuring this and providing any indications. This needs to be used carefully.

-> The Electron Localization function (ELF) has been utilized to discuss the lone-pairs in the structure. However, the results are not clear. I cannot find any difference between Figures 2e and 2f, corresponding in two different values of pressure. Unless I am missing something, the two figures look identical (is that possible?). Have the authors considered to apply a more directional analyses based on their ELF data? For example, plotting the ELF line profile between two certain atoms inside the

structure, one can extract useful information about the chemical bonding (covalent bonds, no bonding, strength of the bond, etc).

-> In Figure 4a what is the difference between the two atomistic models? The pressure is different but the structures look identical. This figure does not provide any useful information.

-> The authors discuss in the main text and SI about vibrations, stretching and bending of bonds, as well as about the dynamical behaviour of the atoms. A very useful, intuitive calculation about these properties is the vibrational density of states (VDOS). Calculating and plotting the VDOS at 300K for the two different pressure values (0 and 9.4 GPa) should provide useful information from the simulation data, and of course illustrating further any arguments.

-> I understand that the authors performed atomistic, DFT simulations (static and molecular dynamics) to provide some insight about their structure. However, the simulations are not very convincing. The model structures are too small. How one can assess the significance of size effects on their findings? In addition, for the calculations of the MSD, the molecular-dynamics simulations are too short. How one can tell if the MSD behaviour remains unchanged if they run a little longer the MD simulation? Moreover, the authors mentioned in the text that the application of pressure is a natural choice for the design of their hybrid structure. However, they used an NVT ensemble for their MD simulations. Why not to run the simulations with an NPT ensemble and let the structure naturally to form/adjust within the framework due to the applied/target pressure? Then, the simulations would have provided a more direct argument.

-> In Table S1, the reported lattice parameters correspond to the experimental data or to the relaxed DFT structures? A comparison is needed.

-> In Figures S11 and S12, why the electronic structure at a pressure of 9.4 GPa is not presented? This value of the pressure seems to be critical from the rest of the results presented in the manuscript, hence it would have been significant to see the density of states (DOS) and investigate the behaviour of the electronic structure.

-> In Table S2, the changes of the Bader charges from 0 to 9.4 GPa pressures are insignificant. One should expect the changes in the atomic structure to be reflected on the Bader ionic charges (difference between crystal/glass).

-> In Figure S13, the three atomistic structures look, again, identical. I cannot see any difference. The figures do not provide any useful information.

Overall, I would like the authors to consider my concerns, comments, questions and suggestions. At the moment, a recommendation for publication in Nature Communications cannot be justified. Nevertheless, I would like to give them the opportunity to address all of them and respond accordingly.

0. It is strange that there is no diffraction from the pressure-transmitting medium (Ne) on the Figures S1 and Figure 1a (expected at pressures higher than 10 GPa).

1. Fig. 2c does not contain standard uncertainties of fitted parameters. One can expect significant correlations between V_0 and B_0 for the high-pressure phase, because V_0 can not be constrained by the experiment. This information should be provided.

2. Pressure-dependent electrical conductivity of $\text{Cu}_{12}\text{Sb}_4\text{S}_{13}$ does not agree with earlier studies of Kitagawa *et al.* (10.7566/JPSJ.84.093701). The Authors should comment on that.

3. It can be noticed that the transition pressure is suspiciously close to Neon crystallization pressure (see Figure 2d). Several authors have pointed out the effect of pressure-transmitting medium incorporation into framework compounds, leading to changes of compressional behavior (*e.g.* Chapman *et al.* 10.1021/ja804079z). A proper study would require a test of different pressure-transmitting media. For example, the Authors could include some diffraction patterns from their experiments where NaCl was used as a pressure-transmitting medium.

To summarize, I am not convinced that the phenomenon, which is described in the manuscript is novel enough to warrant the publication in Nature Communications. Nevertheless, if technical flaws are properly handled, the paper will definitely fit a high-profile specialized journal (*e.g.* PRL, or Chem. Mater).

Point-by-point Responses to the Reviewers' Comments

(Manuscript ID: NCOMMS-21-44547-T)

We are very grateful to the reviewers for their time and efforts to review our manuscript. Both reviewers' comments are encouraging and very helpful for further improving the quality of our paper. We believe that the following responses have addressed all reviewers' concerns thoroughly.

Reviewer #1

Bu *et al.* report the formation of a nested order-disorder framework in $\text{Cu}_{12}\text{Sb}_4\text{S}_{13}$ compound, induced by an application of high pressure. The system was thoroughly studied by a range of experimental and theoretical methods including synchrotron X-ray diffraction, Raman and UV-Vis-NIR spectroscopy, first-principles molecular dynamics simulations, *etc.* The authors claim that they have discovered a new category of solid-state materials, constructed of a crystalline matrix with self-filled amorphous innards. I disagree with this statement. First, the formation of amorphous guest in a crystalline host matrix is a well-known phenomenon in high-pressure metals (see, for example, McMahon *et al.* 10.1103/PhysRevLett.93.055501, McBride *et al.* 10.1103/PhysRevB.91.144111). Although, 'melted' chains in high-pressure metals are not nested, there is a clear similarity of this phenomenon with the disorder of $\text{Cu}_{12}\text{Sb}_4\text{S}_{13}$ substructure. Unfortunately, the Authors do not even mention this in their manuscript. Metal-organic or metal-inorganic frameworks with strongly disordered guest molecules are also not uncommon in scientific literature.

Response: We appreciate the reviewer's insightful comments and the corresponding revisions have been made in the revised manuscript. We have added two new sets of experimental data including *in situ* X-ray absorption spectroscopy (XAS, Figures 3e-f and S6) and scanning transmission electron microscopy (STEM, Figures 3a-d and S5). These

experimental evidence and the corresponding discussion have been added to support our claim in the revised manuscript.

Here, we would like to emphasize the unique features of the nested order-disorder framework (NOF) and its differences in comparison to the cases of ‘melted’ chains in high-pressure metals and metal-organic or metal-inorganic frameworks. There are three unique features in this reported NOF structure, (1) the hybridization of crystalline and amorphous structure is realized within one unit cell in a single compound; (2) the retained crystalline matrix possesses long-range periodicity; (3) the structural modulation occurs at the sublattice-scale level due to the chemical-bond hierarchy.

On the one hand, the chain “melting” occurs in the metals who possess the incommensurate host-guest structure (Figure R1a).¹⁻³ At certain pressure and temperature, the long-range ordering of the guest chains is lost while the host atoms remain crystalline (Figure R1b), resulting the order-disorder hybrid structure. The melted chain in high-pressure metals is one of the most exciting discoveries in the exploration of order-disorder hybrid materials, which have been included in the revised version of Introduction (on page 2 of the revised manuscript). We agree with the reviewer that the formation of amorphous guest in a crystalline host in the melted chains in high-pressure metals is similar with the reported NOF structure. Here, we would like to state their differences from the following two aspects. (1) The NOF structure is realized within one unit cell, while the chain “melting” happens in higher dimensions and the initial unit cell does not retain. (2) The disordering of the sublattice in NOF structure takes place in three directions due to the chemical-bond hierarchy, while the order-disorder transition in the guest chains takes place in one direction caused by the incommensurate structure. Therefore, the NOF structure reported in our work is distinguished from the melted chains in incommensurate metals.

On the other hand, metal-organic or metal-inorganic frameworks with disordered guest

molecules consist of two individual compounds that no chemical bonding is formed between them,^{4, 5} such as gas-MOF structures (Figure R1c);⁶ while the NOF structure is realized within a single compound with chemical bonding hierarchy where the amorphous innards are self-filled in the periodically ordered crystalline matrix (Figure R1d). The MOF-related structures do not fit the features #1 and #3 of NOF mentioned above. Therefore, the NOF structure reported in this work is novel which is distinguished from the cases of ‘melted’ chains in some metals and metal-organic or metal-inorganic frameworks. We would like to express our sincere gratitude to the reviewer again for the insightful comment. Carefully considering the similarities and differences between these distinguishable hybrid systems is thus beneficial for the deeper understanding of each of them.

Figure R1. (a) Basic host–guest motif of the K-IIIa crystal structures. The red lines indicate the basic host unit cell.² (b) Part of integrated profiles of Rb-IV under pressure. (1001) is the strongest reflection from the guest component; other reflections are from the host component.¹ (c) A representation of In-MOF structure highlighting the distinct guest-accessible cavities.⁶ (d) The increasing anharmonic Cu vibration during compression causes a disordered Cu2 sublattice embedded in the retained Cu1 crystalline matrix,

forming the NOF.

The description of 'melted' chains in high-pressure metals and MOF structures have been included in the introduction of the revised manuscript.

On page 2: "These hybrid materials could possess advantageous properties from both crystalline and disordered units, which are increasingly attractive for potential technological applications, including black TiO₂ nanomaterials for photocatalysis,^{5, 6} two-dimensional electron gases at crystalline-amorphous oxide interfaces for transparent conductors,⁷ metal-organic frameworks (MOFs) and their composites for catalysis.⁸ From the local structure point of view, hybridization has been made at the mesoscopic scale,⁹⁻¹¹ such as paracrystalline silicon,¹² intermediate crystalline metallic glass,¹³ and melted chains in high-pressure metals.¹⁴⁻¹⁶"

Apart from that there are several issues with the data presentation, which I would like to point out:

1. On Page 4 the Authors write that the structural softening "is related to the large atom displacement parameter of Cu₂". In the experimental section the Authors write that Rietveld refinement was performed (Figure S1). Therefore, it should be possible to track the development of the refined Cu₂ ADP with pressure increase. This information should be given in the text.

Response: We thank the reviewer for the helpful suggestion. The atomic displacement parameters (ADP) of Cu₂ atoms have been added in the Supplementary Information. The values are derived from the Rietveld refinement of synchrotron X-ray diffraction experiments. The ADP of Cu₂ increases as the pressure increases (Figure S2), indicating the enhanced rattling vibration. Eventually, the large rattling of Cu₂ converts to be diffusing when pressure exceeds 10 GPa and thus contributes to disordered Cu₂

sublattice and structural softening.

The corresponding description has been added in the Revised Manuscript:

On pages 5 and S14: “ As shown in Figure S2, the ADP values of Cu2 significantly increase from 0.01 to 0.06 Å² during compression, indicating the enhanced rattling vibration and the moveable Cu2 atoms under high pressures.”

Figure S2. Cu2 atomic displacement parameters (ADP) of Cu₁₂Sb₄S₁₃ at different pressures. The ADP values are derived from the Rietveld refinement of synchrotron X-ray diffraction.

2. Following the previous comment, if Rietveld refinement was performed at all pressure points, the corresponding cif files must be provided in the Supplementary Information.

Response: Thanks for pointing out this. The cif files at different pressures have been submitted as Supplementary Information.

3. How was the disorder modelled in the Rietveld refinements at high pressures (Figure S1, 13.3 and 16.9 GPa)?

Response: Rietveld refinement is used for the characterization of crystalline materials and

the information about defects and disorders is hidden in the background of a powder XRD pattern (Figure R2). During compression, the increased diffuse background provides qualitative information about the increasing disordered states. For the Rietveld refinements of the XRD patterns at 13.3 and 16.9 GPa, we took the broad diffuse background as the baseline, which was widely used previously.⁷⁻⁹ For the XRD patterns at high pressures, the main sharp diffraction peaks are corresponding to the rigid Cu1 crystalline framework, while the broad diffuse background and the significantly weakened peaks are related to partially disordered Cu2 subunit. In the Rietveld refinements, we deducted the broad diffuse background and used the same crystalline model (*I-43m*) to refine the structures at 13.3 and 16.9 GPa, which shows low values of R_{wp} , R_p , and R_F parameters in the final Rietveld refinements. For the investigation of disordered structures, we further performed the *in situ* high pressure near edge region of X-ray absorption spectra and the extended X-ray absorption fine structure at Cu K-edge to explore the electronic occupation states and atomic distances around Cu atoms.

Figure has been redacted.

Figure R2. The information contents of a powder XRD pattern.

In the revised manuscript, *in situ* high-pressure X-ray absorption spectra (XAS) at Cu K-

edge were collected to investigate the evolution of the coordination environment of the Cu atoms (Figures 3e and S6a). In the analysis of the near edge region of XAS, the Cu ions exhibit mixed-valences of Cu^+ and Cu^{2+} in $\text{Cu}_{12}\text{Sb}_4\text{S}_{13}$ (Figure 3e). The obviously weakened peak at around 8983 eV beyond 8.9 GPa implies the weakening of the Cu₂-S bonding. Figures 3f and S6b show the Fourier transform (FT) plots and *k*-weight of the extended X-ray absorption fine structure (EXAFS), respectively. The distances around 1.5–2.3 Å are considered as common Cu-S covalent bonds (Figure 3f).^{10, 11} The obviously weakened and broadened Cu-S peaks beyond 8.9 GPa indicate the widely distributed Cu-S bond lengths and complex coordination environment of Cu₂ atoms, which confirms the distorted innards in NOF structure.

In the revised manuscript, *in situ* XAS results with detailed discussion have been added to address the reviewer's concern.

On pages 7 and 8:

“ Furthermore, *in situ* high-pressure X-ray absorption spectra (XAS) were collected to understand the evolution of atomic coordination environment (Figures 3e and S6a). In the analysis of the near edge region of XAS, the Cu ions exhibit mixed-valences of Cu^+ and Cu^{2+} in $\text{Cu}_{12}\text{Sb}_4\text{S}_{13}$ (Figure 3e). The obviously weakened peak at around 8983 eV beyond 8.9 GPa implies the weakening of the Cu₂-S bonding. Figures 3f and S6b show the Fourier transform (FT) plots and *k*-weight of the extended X-ray absorption fine structure (EXAFS), respectively. The distances around 1.5–2.3 Å are considered as common Cu-S covalent bonds (Figure 3f).^{36, 37} The obviously weakened and broadened Cu-S peaks beyond 8.9 GPa indicate the widely distributed Cu-S bond lengths and complex coordination environment of Cu₂ atoms, which confirms the distorted innards in NOF structure.”

Figures 3. Atomic coordination environment of $\text{Cu}_{12}\text{Sb}_4\text{S}_{13}$. (e) The near edge region of XAS on the K -edge of Cu for $\text{Cu}_{12}\text{Sb}_4\text{S}_{13}$ at different pressures. (f) FT curves of the EXAFS data of $\text{Cu}_{12}\text{Sb}_4\text{S}_{13}$.

On page S5 of the Supplementary Information: **“X-ray absorption spectra measurement.** *In situ* high-pressure X-ray absorption spectra (XAS) of Cu K -edge were collected at BL05U station in Shanghai Synchrotron Radiation Facility (SSRF). The energy dispersive mode was used for studies of materials under high pressure. To avoid DACs glitches, polycrystalline diamond anvils were used for XAS measurements under pressure. The XAS data of the samples were collected at different pressure in transmission mode. Internal energy calibration was accomplished by measuring the standard Cu foil. The acquired XAS data were processed according to the standard procedures using the Athena implemented in the IFEFFIT software packages.⁹”

On page S19:

Figure S6. (a) The raw high-pressure *in situ* XAS data of $\text{Cu}_{12}\text{Sb}_4\text{S}_{13}$ on the K -edge of Cu under pressure. (b) The k -weighted EXAFS spectra for $\text{Cu}_{12}\text{Sb}_4\text{S}_{13}$ at different pressures.

4. It is strange that there is no diffraction from the pressure-transmitting medium (Ne) on the Figures S1 and Figure 1a (expected at pressures higher than 10 GPa).

Response: We would like to note that silicon oil was used as the pressure-transmitting medium in the single-crystal XRD experiments, thus no diffraction from neon can be observed at above 10 GPa. Figures R3a and R3b show the detailed XRD image and integrated patterns of $\text{Cu}_{12}\text{Sb}_4\text{S}_{13}$ beyond 10 GPa, respectively. On the other hand, neon was used as a pressure transmitting medium in the powder XRD experiments. As shown in Figures R3c and R3d, the diffraction of neon appears at 7.2 GPa and reaches a relatively high intensity at 9.4 GPa. In order to better refine the powder XRD data using the Rietveld method, we masked the neon diffraction spots using Dioptas software.¹² That's why Figure S1, showing the Rietveld refinement results, doesn't show the diffraction from neon. We thank the reviewer for pointing out this and the information about the pressure transmitting medium for each high-pressure experiment has been included in the revised supplementary information.

On page S4 of Supplementary Information: “Silicon oil was used as the pressure transmitting medium in the single-crystal XRD experiments.” and “Neon was used as the pressure transmitting medium in the powder XRD experiments.”

Figure R3. (a) The single-crystal XRD image of $\text{Cu}_{12}\text{Sb}_4\text{S}_{13}$ at 10.7 GPa, and (b) the integrated XRD patterns at 10.7 and 12.2 GPa. (c) The powder XRD image of $\text{Cu}_{12}\text{Sb}_4\text{S}_{13}$ at 7.2 GPa and (d) the integrated XRD patterns at 7.2 and 9.4 GPa. The diffraction of neon appears at 7.2 GPa and strengthens at 9.4 GPa.

5. Fig. 2c does not contain standard uncertainties of fitted parameters. One can expect significant correlations between V_0 and B_0 for the high-pressure phase, because V_0 cannot be constrained by the experiment. This information should be provided.

Response: Thanks for pointing out this and the standard uncertainties of fitted

parameters have been included in the revised Figure 2c of the revised manuscript.

On page 6:

Figure 2c. Unit-cell volume during compression. The values of bulk modulus B_0 were determined to be 56.5(7) and 36.9(2) GPa in the low-pressure (LP) and high-pressure (HP) regions, respectively.

6. Pressure-dependent electrical conductivity of $\text{Cu}_{12}\text{Sb}_4\text{S}_{13}$ does not agree with earlier studies of Kitagawa *et al.* (10.7566/JPSJ.84.093701).¹³ The Authors should comment on that.

Response: We thank the reviewer for this comment. At room temperature, the variation behavior of the electrical conductivity from our work is consistent with the one reported by Kitagawa *et al.*,¹³ which shows decreasing resistivity during compression from 0 to 4 GPa (Figures R4a and R4b). The difference lies in the temperature-dependent conductivity under pressures. The ρ - T curve of the sample in Kitagawa's study, for example at 4.06 GPa, shows a turning point at 160 K, below which the sample exhibits metallic behavior (the orange curve in Figure R4a). Although the ρ - T curve of our sample at 5.2 GPa shows a kink at around 160 K, it retains the semiconductive behavior (Figure R4c). The different behavior may be due to the different sample preparation methods. We used the solvent-thermal method to synthesize the single crystals at a relatively low temperature of 200 °C.

The lattice constant of the single crystal is determined to be $a=10.31 \text{ \AA}$ using single-crystal analysis at ambient condition (Figure R5) which is in line with the reported structure of high-quality single-crystal $\text{Cu}_{12}\text{Sb}_4\text{S}_{13}$.¹⁴ While the study by Kitagawa used a powder sample which was prepared using solid-state reactions at relatively high temperature of $900 \text{ }^\circ\text{C}$, which contains a small amount of CuSbS_2 as a byproduct.¹³ In addition, such a high synthetic temperature may produce some Cu vacancies in $\text{Cu}_{12}\text{Sb}_4\text{S}_{13}$ compound,^{15, 16} together with the CuSbS_2 impurity, they could be responsible for the difference.

Figure R4. (a) Temperature-dependent electrical resistivity (ρ - T) at different pressures reported by Kitagawa *et al.*¹³ Note that their powder sample is identified to be $\text{Cu}_{12}\text{Sb}_4\text{S}_{13}$ with a small amount of CuSbS_2 . (b)-(c) The electrical resistivities of our single-crystal $\text{Cu}_{12}\text{Sb}_4\text{S}_{13}$ at different pressures.

Figure R5. Indexing results of the single-crystal XRD for $\text{Cu}_{12}\text{Sb}_4\text{S}_{13}$ at ambient condition.

7. It can be noticed that the transition pressure is suspiciously close to Neon crystallization pressure (see Figure 2d). Several authors have pointed out the effect of pressure-transmitting medium incorporation into framework compounds, leading to changes of compressional behavior (*e.g.* Chapman *et al.* 10.1021/ja804079z). A proper study would require a test of different pressure-transmitting media. For example, the Authors could include some diffraction patterns from their experiments where NaCl was used as a pressure-transmitting medium.

Response: We thank the reviewer for this comment. As has been stated in the response to comment 4, we used different pressure transmitting media, neon, and silicon oil, for the XRD experiments. As shown in Figure R6a, the XRD results obtained using different pressure transmitting media show the consistent compressional behavior, indicating that neon doesn't affect the structural variations. In addition, the cavity in $\text{Cu}_{12}\text{Sb}_4\text{S}_{13}$ is too small (3.54 Å) for the neon molecular (4.46 Å) to incorporate in (Figure R6b).¹⁷ While for the work reported by Chapman *et al.* 10.1021/ja804079z,⁴ they used MOF structures that possess large cavities usually over 5 Å, which is big enough for neon to incorporate in and the effect of neon crystallization should be carefully considered in that case.

Figure R6. (a) Bond angles of S–Sb–S as a function of pressure. The results are derived from the XRD data collected using different pressure transmitting media of neon and silicon oil. (b) Crystal structure of $\text{Cu}_{12}\text{Sb}_4\text{S}_{13}$ where the diameter of the cavity is about 3.54 Å.

Reviewer #2

In this manuscript, Bu et al. report the design of a nested order/disorder copper chalcogenide ($\text{Cu}_{12}\text{Sb}_4\text{S}_{13}$) structure via the application of external high pressure. The concept of the study is very interesting and the objective of the quest of a hybrid crystalline/amorphous material, which combines the features of both states simultaneously in a structure, is timely and of significant importance.

The authors have used a range of experimental techniques, which are described in details in the Supplementary Information. In addition, density-functional theory (DFT) simulations have been used to provide some theoretical evidence. Moreover, the authors performed some extensive analyses, and from different perspectives, to examine their experimental and simulation data in order to verify their view. Therefore, it is important to highlight and appreciate their efforts as reported in this study.

My main concern based on the results is how valid is the claim of the realization of this hybrid amorphous/crystalline material. A structure with increased disorder does not necessarily mean that it is a glassy structure. For example, the cubic (rocksalt) crystalline structure of the chalcogenide phase-change memory material, $\text{Ge}_2\text{Sb}_2\text{Te}_5$, is a highly-disordered crystalline structure. Such a structure, does not have any resemblance with an amorphous, glass-like structure. Another example are the grain boundaries, where two different crystalline structures, in a polycrystalline material, form a specific defective interface.

Response: We appreciate the reviewer's very positive comments and the insightful conclusion of our work. To address the reviewer's concern, we have conducted more experiments, added two new sets of experimental data including *in situ* X-ray absorption spectroscopy (XAS) and scanning transmission electron microscopy (STEM). These experimental evidence and the corresponding discussion have been added to support our

claim in the revised manuscript.

Here, we would like to emphasize the unique features of the nested order-disorder framework (NOF) which make it different from the cases of highly-disordered crystalline structure and grain boundaries. In brief, the reported NOF structure possesses three unique features, (1) the hybridization of crystalline and amorphous structure is realized within one unit cell in a single compound; (2) the retained crystalline matrix possesses long-range periodicity; (3) the structural modulation occurs at the sublattice-scale level due to the chemical-bond hierarchy.

The structure of retained crystalline matrix can be elucidated by the single-crystal XRD which has been discussed in the original manuscript. To further investigate the disordered subunit of the NOF structure, *in situ* high-pressure X-ray absorption spectra (XAS) at Cu K-edge were collected to investigate the variations of the coordination environment of the Cu atoms (Figures 3e and S6a). In the analysis of the near edge region of XAS, the Cu ions exhibit mixed-valences of Cu⁺ and Cu²⁺ in Cu₁₂Sb₄S₁₃ (Figure 3e). The obviously weakened peak at around 8983 eV beyond 8.9 GPa implies the weakening of the Cu–S bonding. Figures 3f and S6b show the Fourier transform (FT) plots and *k*-weight of the extended X-ray absorption fine structure (EXAFS), respectively. The distances around 1.5–2.3 Å are considered as common Cu–S covalent bonds (Figure 3f).^{10, 11} The obviously weakened and broadened Cu–S peaks beyond 8.9 GPa indicate the widely distributed Cu–S bond lengths and complex coordination environment of Cu₂ atoms, which confirms the distorted innards in NOF structure. Thus, we would like to conclude that the amorphous innards in the NOF structure is somewhat similar to the highly-disordered crystalline structures, where the disordered states are inherited from the original crystalline lattices.¹⁸

In the revised manuscript, the *in situ* XAS results and the corresponding discussion have been included.

On pages 7 and 8: “ Furthermore, *in situ* high-pressure X-ray absorption spectra (XAS) were collected to understand the evolution of atomic coordination environment (Figures 3e and S6a). In the analysis of the near edge region of XAS, the Cu ions exhibit mixed-valences of Cu⁺ and Cu²⁺ in Cu₁₂Sb₄S₁₃ (Figure 3e). The obviously weakened peak at around 8983 eV beyond 8.9 GPa implies the weakening of the Cu–S bonding. Figures 3f and S6b show the Fourier transform (FT) plots and *k*-weight of the extended X-ray absorption fine structure (EXAFS), respectively. The distances around 1.5–2.3 Å are considered as common Cu–S covalent bonds (Figure 3f).^{36, 37} The obviously weakened and broadened Cu–S peaks beyond 8.9 GPa indicate the widely distributed Cu–S bond lengths and complex coordination environment of Cu₂ atoms, which confirms the distorted inroads in NOF structure.”

Figures 3. Atomic coordination environment of Cu₁₂Sb₄S₁₃. (e) The near edge region of XAS on the *K*-edge of Cu for Cu₁₂Sb₄S₁₃ under pressure. (f) FT curves of the EXAFS data of Cu₁₂Sb₄S₁₃.

On page S5 of the Supplementary Information: “**X-ray absorption spectra measurement.** *In situ* high-pressure X-ray absorption spectra (XAS) of Cu *K*-edge were collected at BL05U

station in Shanghai Synchrotron Radiation Facility (SSRF). The energy dispersive mode was used for studies of materials under high pressure. To avoid DACs glitches, polycrystalline diamond anvils were used for XAS measurements under pressure. The XAS data of the samples were collected at different pressure in transmission mode. Internal energy calibration was accomplished by measuring the standard Cu foil. The acquired XAS data were processed according to the standard procedures using the Athena implemented in the IFEFFIT software packages.⁹

On page S19:

Figure S6. (a) The raw high-pressure XAS data of $\text{Cu}_{12}\text{Sb}_4\text{S}_{13}$ on the K -edge of Cu under pressure. (b) The k -weighted EXAFS spectra for $\text{Cu}_{12}\text{Sb}_4\text{S}_{13}$ as a function of pressure.

Throughout the manuscript there are also some certain things that require further clarification:

1. I think the Introduction is poor. There is only one paragraph where the authors are trying to build up the objective. I would prefer, as a reader, to get more convinced about the importance of the design of such hybrid structures. For instance, there are no

applications mentioned of these materials. How one can utilize their enhanced properties in specific applications? Why it is important to tailor and exploit the features of both states regarding the operation of devices? Tackling these questions should strengthen the arguments of the study.

Response: Thanks so much for the insightful suggestion. The Introduction section of the revised manuscript has been carefully revised which includes the importance of the designed NOF structure, more examples of crystalline-amorphous hybrid structures and the potential applications of these materials.

On page 2: “These hybrid materials could possess advantageous properties from both crystalline and disordered units, which are increasingly attractive for potential technological applications, including black TiO₂ nanomaterials for photocatalysis,^{5, 6} two-dimensional electron gases at crystalline-amorphous oxide interfaces for transparent conductors,⁷ metal-organic frameworks (MOFs) and their composites for catalysis.⁸ From the local structure point of view, hybridization has been made at the mesoscopic scale,⁹⁻¹¹ such as paracrystalline silicon,¹² intermediate crystalline metallic glass,¹³ and melted chains in high-pressure metals.¹⁴⁻¹⁶”

On pages 2 and 3: “ Due to variable coordination conditions and valence states, copper chalcogenides have large structural variability and exhibit an intrinsic chemical-bond hierarchy, which gives a high and anisotropic tunability.¹⁸⁻²⁰ Besides chemical tailoring, the degree of bonding hierarchy can be tuned by applying external stimuli, including temperature, pressure, and electric field.^{17, 18} Recently, temperature-induced hybrid state has been reported in Cu₂Se where the Cu⁺ sublattice becomes amorphous on warming and induced liquid-like flow.^{19, 21} Besides, the amorphous-to-crystal transition can be triggered by electric pulses in phase-change memory material Ge₂Sb₂Te₅ with bonding energy hierarchy.^{17, 22} However, strong vibration of all atoms at high temperature or electric field leads to the whole structural instability and second-phase precipitation,

which limits the tunability and formation of crystalline-amorphous hybrid structures.²³ As a state variable, pressure provides an effective and clean approach to adjust the atomic interactions and thus alter the bonding configuration without changing chemical compositions.²⁴⁻²⁷ Therefore, pressure processing enables the exploration and modulation of crystalline-amorphous hybrid structures which would collaboratively optimize the competing physical properties.”

2. The authors use the terminology of a void, which is a specific term in amorphous materials to define certain space. Also, certain rules (e.g. Voronoi polyhedra, and others) are typically employed to calculate and characterize voids. The term is used rather loosely in the manuscript and it needs either justification or revision.

Response: Thanks for pointing out the inappropriate use of “void”. It has been changed to “space” in the revised manuscript on Page 3.

On page 3: “Each Sb atom is bonded to three S atoms, giving a space in the structures occupied by the lone pair electrons (LPEs).”

3. Pressure was utilized to create the hybrid structure. Such choice makes sense and I do not have any objection. In modern literature has been demonstrated that the application of an external electric field can modify the atomic and electronic structures of crystalline and amorphous materials. Have the authors considered such an option? It should be discussed as means of modifying structures and tailoring properties, at least.

Response: We thank the reviewer for such a good suggestion. The external electric field can modify the atomic structures of crystalline and amorphous materials. A well-known example is the phase-change memory material $\text{Ge}_2\text{Sb}_2\text{Te}_5$ (GST).^{19, 20} It changes from a

covalently bonded amorphous phase to a resonantly bonded metastable cubic crystalline phase under electric pulses.²⁰ The basic principle is to take advantage of the property contrast between the crystalline and amorphous states to encode information.¹⁹ Recent report has demonstrated that distortions in GST crystals that have a bonding energy hierarchy trigger the destruction of the subsystem of weaker bonds and subsequent collapse of the long-range order, generating the amorphous phase.¹⁸ Such a process is similar to the forming of amorphous innards in our case. The difference is that our case forms a crystalline-amorphous hybrid NOF structure instead of a highly-disordered structure. The hybrid NOF structure combines advantages from both crystalline and amorphous subunits and realizes the collaborative improvement of two competing properties (low thermal and high electrical conductivity). In the revised introduction, we have added the corresponding description about the external electric field and state why we chose pressure to exploit the crystalline-amorphous hybrid structures.

The corresponding discussion has been added on page 2 of the revised manuscript.

“Besides chemical tailoring, the degree of bonding hierarchy can be tuned by applying external stimuli, including temperature, pressure, and electric field.^{17, 18} Recently, temperature-induced hybrid state has been reported in Cu_2Se where the Cu^+ sublattice becomes amorphous on warming and induced liquid-like flow.^{19, 21} Besides, the amorphous-to-crystal transition can be triggered by electric pulses in phase-change memory material $\text{Ge}_2\text{Sb}_2\text{Te}_5$ with bonding energy hierarchy.^{17, 22} However, strong vibration of all atoms at high temperature or electric field leads to the whole structural instability and second-phase precipitation, which limits the tunability and formation of crystalline-amorphous hybrid structures.²³ As a state variable, pressure provides an effective and clean approach to adjust the atomic interactions and thus alter the bonding configuration without changing chemical compositions.²⁴⁻²⁷ Therefore, pressure processing enables the exploration and modulation of crystalline-amorphous hybrid

structures which would collaboratively optimize the competing physical properties.”

4. The NOF structure reported in Figure 1b is it from real data (trajectory/positions) or it is a sketch/graphic? It is not clear. If it corresponds to the simulation trajectory, from the MD simulations, then the argument is more valid. In contrast, if it is a sketch I am afraid it is not enough. It is also rather blurred.

Response: Thanks for the comment. We would like to note that it is a schematic illustration to show the features of the NOF structure. To address the reviewer’s concern, we have further performed atomic-resolved scanning transmission electron microscopy (STEM) and extended the first-principles simulations. In the revised version, we modified the schematic illustration (Figure 1b). Compared to the previous graphic, it is more accurate to describe the disordered Cu2 atoms diffusing within the Cu1 crystalline framework rather than in the whole crystalline structure. Such behavior can be demonstrated by both the experimental STEM results and the first-principles molecular dynamic simulations. The detailed discussion can be seen as follows.

On page 4:

Figure 1b. Pressure-induced transformation of crystalline $\text{Cu}_{12}\text{Sb}_4\text{S}_{13}$ to the NOF structure. The increasing anharmonic Cu vibration during compression causes a disordered Cu2

sublattice embedded in the retained Cu1 crystalline matrix, creating the NOF.

In the experiment, we added spherical aberration-corrected STEM images that directly show a NOF structure. Figures 3a and S5 show the high-angle annular dark-field (HAADF) STEM images of $\text{Cu}_{12}\text{Sb}_4\text{S}_{13}$ projected along the $[1\ 1\ -2]$ and $[0\ 2\ -1]$ zone-axis at ambient condition, with neighboring atom columns of Sb/Cu and Cu1/Cu2. Due to the strong scattering electron ability of Sb atoms, the two atom columns clearly show weak and strong contrast in Z-contrast HAADF (Figures 3a and 3b). After the high-pressure treatment, the disordering occurs within the $(1\ 1\ 1)$ planes, which is related to the Cu2 atoms (top panel of Figure 3a and Figure 3b). From the intensity-scan profiles, one can see a good periodicity for the initial sample (Figure 3c). While after high-pressure treatment, the background lifts and additional peaks corresponding to Cu2 atoms can be observed in adjacent Sb/Cu atom columns (Figure 3d). This observation indicates that Cu2 atoms move away from the equilibrium position and become disordered in the crystalline matrix. Moreover, as shown in the select area electron diffraction (inset of Figure 3b), the retained diffraction spots corresponding to the structure of Cu1 framework sit on the broad diffuse background, which is consistent with the XRD results in Figure 2, confirming the formation of NOF structure.

In first-principles molecular dynamic (FPMD) simulations (Figure 5), the trajectory of Cu atoms in $\text{Cu}_{12}\text{Sb}_4\text{S}_{13}$ is greatly promoted under high pressures and the mean square displacement (MSD) values of Cu2 are higher than the Lindemann melting threshold, indicating the existence of liquid-like Cu2 sublattice under high pressure.²¹ Based on these experimental and simulation results, we draw a schematic illustration to show the liquid-like Cu2 cations diffusing in the rigid Cu1 crystalline framework, forming the NOF structures.

More details about the spherical aberration-corrected STEM images have been added in the revised version of the manuscript and Supplementary Information (Figure 3 on Page

7, Figure S5 on Page S18).

On pages 7 and 8:

“To investigate the local structure of NOF structure, we examined the atomic-scale structures of pristine and high-pressure treated samples using spherical aberration-corrected scanning transmission electron microscopy (STEM). Figures 3a and S5 show the high-angle annular dark-field (HAADF) STEM images of $\text{Cu}_{12}\text{Sb}_4\text{S}_{13}$ projected along the $[1\ 1\ -2]$ and $[0\ 2\ -1]$ zone-axis at ambient condition, with neighboring atom columns of Sb/Cu and Cu1/Cu2. After the high-pressure treatment, the disordering occurs within the $(1\ 1\ 1)$ planes, which is related to the Cu2 atoms (Figure 3b). From the intensity-scan profiles, one can see a good periodicity for the initial sample (Figure 3c). While after high-pressure treatment, the background lifts and additional peaks can be observed which corresponding to the randomly occupied Cu2 atoms (Figure 3d). This observation indicates that Cu2 atoms move away from the equilibrium position and become disordered in the crystalline matrix. Moreover, as shown in the select area electron diffraction (inset of Figure 3b), the retained diffraction spots corresponding to the structure of Cu1 framework sit on the broad diffuse background, which is consistent with the XRD results in Figure 2, confirming the formation of NOF structure.”

Figure 3. Atomic-scale analysis of $\text{Cu}_{12}\text{Sb}_4\text{S}_{13}$. The HAADF STEM images taken along the $[1\ 1\ -2]$ zone-axis of (a) the initial sample at ambient condition and (b) the sample treated by high pressure. The top panel of (a) shows the $(1\ 1\ 1)$ planes that are related to the Cu2 atoms. The inset of (b) shows the corresponding select area electron diffraction. The retained electrons diffraction spots with the broad diffuse background confirm the hybrid structure. Intensity-scan profiles taken from the (111) plane with Sb/Cu atom columns of (c) the initial sample at ambient and (d) the sample treated by high pressure, as indicated by a yellow arrow in (a) and (b).

On page S5: **“Transmission electron microscopy.** The pristine and high-pressure treated samples were examined. The scanning transmission electron microscopy (STEM) specimens were prepared by focused ion beam (FIB, Helios nanolab 600, FEI, USA). The atomic-scale high-angle annular dark-field (HAADF) STEM images were carried out on a spherical aberration-corrected Hitachi HF5000 operating at 200 kV.”

On page S18:

Figure S5. The HAADF image taken at ambient conditions along the [0 2 -1] zone-axis. (Inset) Select area electron diffraction from this region.

5. The authors mention in the text softening of the structure without really measuring this and providing any indications. This needs to be used carefully.

Response: We thank the reviewer for the suggestion. We agree that “more compressible” should be more accurate to describe the situation of the decreased bulk modulus under high pressures. The bulk modulus B_0 fitted from the Birch-Murnaghan equation can quantitatively provide the compressibility of the solids.²² Generally, materials become hard to compress due to a decrease of the interatomic spacing, which shows higher B_0 after structural transition under pressure. However, $\text{Cu}_{12}\text{Sb}_4\text{S}_{13}$ shows abnormally decreased B_0 from 56.5(7) to 36.9(2) GPa before and after structural transition, indicating more compressible structure due to the movable Cu_2 amorphous innards in NOF. Corresponding change has been made in the revised manuscript.

On page 5: “Such an abnormal decrease of B_0 indicates the more compressible structure under high pressures”.

6. The Electron Localization function (ELF) has been utilized to discuss the lone-pairs in the structure. However, the results are not clear. I cannot find any difference between Figures 2e and 2f, corresponding in two different values of pressure. Unless I am missing something, the two figures look identical (is that possible?). Have the authors considered to apply a more directional analyses based on their ELF data? For example, plotting the ELF line profile between two certain atoms inside the structure, one can extract useful information about the chemical bonding (covalent bonds, no bonding, strength of the bond, etc).

Response: We thank the reviewer for such a good suggestion. We plotted the ELF line profile between Sb and S atoms inside the structure as the new Figure 2f, which clearly shows the suppressed lone pair electrons (LPEs) under pressure. While the valence electron density maps (original Figure 2f) have been moved into Supplementary Information.

On page 6:

Figure 2. (e) The ELF graphs and (f) the ELF line profiles between Sb and S atoms at ambient pressure and at 9.4 GPa.

On page S17:

Figure S4. Valence electron density maps at ambient pressure and at 9.4 GPa. The ball-and-stick models with an isosurface value of ELF = 0.94 projects onto the (101) plane.

7. In Figure 4a what is the difference between the two atomistic models? The pressure is different but the structures look identical. This figure does not provide any useful information.

Response: We thank the reviewer for this comment. Figure 4a shows the trajectory of Cu atoms at 0 and 9.4 GPa from molecular dynamic (MD) simulations which indicates the enhanced anharmonic motion of Cu₂ atoms at high pressure. We agree with the reviewer that the difference is not obvious in the large super cell we used. In the revised manuscript, we have zoomed in the trajectory of typical Cu₂ atoms, which shows the enhanced anharmonic motion of Cu₂ atoms more clearly (Figure 5a). Furthermore, the mean square displacement (MSD) line profiles in the time domain were obtained from the MD trajectory (Figure 5b), which indicate the MSD value of Cu₂ atoms surpassing the melting threshold.

Figure 4a about the MD trajectory has been modified in the revised manuscript.

On page 12:

Figure 5. (a) The trajectory of Cu atoms at 0 GPa and 9.4 GPa at room temperature.

8. The authors discuss in the main text and SI about vibrations, stretching and bending of

bonds, as well as about the dynamical behaviour of the atoms. A very useful, intuitive calculation about these properties is the vibrational density of states (VDOS). Calculating and plotting the VDOS at 300K for the two different pressure values (0 and 9.4 GPa) should provide useful information from the simulation data, and of course illustrating further any arguments.

Response: Thank you for the suggestion of introducing VDOS. We have calculated and plotted the VDOS at 300 K at 0 and 9.4 GPa, as shown in Figure S17. The calculation was performed after cell relaxation using NPT ensemble and extended simulation time (10 ps). The low-lying modes ($<100\text{ cm}^{-1}$) of VDOS are mainly attributed to Cu₂ atoms, which signify weak bonding due to their low frequencies.²³ The other atoms contribute to the higher-energy modes ($>100\text{ cm}^{-1}$) and shift towards higher frequencies with pressure increasing, suggesting more rigid bond formation under higher pressures.²³ Whereas, the low-lying modes corresponding to Cu₂ atoms still stay at the low frequencies, which indicate the retaining weak bonding under high pressures. During compression, such bonding hierarchy triggers the destruction of the sublattice with Cu₂ weak bonds but the rest crystalline framework can be well-identified, generating the hybrid NOF structures. The discussion of VDOS has been added into the revised manuscript and the VDOS results have been added in the Supplementary Information (Figure S17).

On page 11: “The vibrational density of states (VDOS) of all atoms at 0 and 9.4 GPa are shown in Figure S17. The low-lying modes ($<100\text{ cm}^{-1}$) are mainly attributed to Cu₂ atoms, which signify weak bonding due to their low frequencies.²⁸ The other atoms contribute to the higher-energy modes and shift towards higher energy with pressure increasing, which suggests more rigid bond formation under pressure.²⁸ Whereas, the low-lying modes corresponding to Cu₂ atoms still stay at the low-frequency region, which retain weak bonding features under high pressures. During compression, the bonding hierarchy induces the destruction of the sublattice with Cu₂ weak bonds but the rest crystalline

framework retains, resulting in the formation of NOF structure.”

On page S32:

Figure S17. Vibrational density of states (VDOS) of $\text{Cu}_{12}\text{Sb}_4\text{S}_{13}$ tetrahedrites at 0 and 9.4 GPa.

9. I understand that the authors performed atomistic, DFT simulations (static and molecular dynamics) to provide some insight about their structure. However, the simulations are not very convincing. The model structures are too small. How one can assess the significance of size effects on their findings? In addition, for the calculations of the MSD, the molecular-dynamics simulations are too short. How one can tell if the MSD behaviour remains unchanged if they run a little longer the MD simulation? Moreover, the authors mentioned in the text that the application of pressure is a natural choice for the design of their hybrid structure. However, they used an NVT ensemble for their MD simulations. Why not to run the simulations with an NPT ensemble and let the structure naturally to form/adjust within the framework due to the applied/target pressure? Then, the simulations would have provided a more direct argument.

Response: We appreciate the reviewer’s comments on the DFT simulations. The reviewer suggested the model structure, which consists of 48 Cu (11 valence electrons), 16 Sb (5 valence electrons), and 52 S (6 valence electrons) atoms (116 atoms), is small. It should

be respectively noted that the total number of electrons of the supercell we used is 920, which is almost at the limit of our simulation power. According to the reviewer's suggestion, we further conducted one set of MD at 0 GPa, 300 K with a double-size simulation box (232 atoms, 1840 electrons, 5000 MD steps) and the results are consistent with the more power-friendly system (Figure R7).

Figure R7. MSD results at (a) 116 atoms system and (b) double-size 232 atoms system.

We agree with the reviewer that adding an NPT step is useful to converge pressure. Along the trajectory, we now initialize simulation with 3 ps NVT simulation for heating, follow by 5 ps NPT to the target pressure, and eventually run 10 ps NVT simulation for equilibrium (previously was 5 ps). Data reduction is taken from the equilibrium run. As shown in Figure R8, the velocity autocorrelation function shows a fully converged system at 0 GPa and 300 K. We would like to note that the updated simulation results are consistent with the previous ones.

Figure R8. Velocity autocorrelation function along the MD trajectory of 0 GPa and 300 K. The system has been in the equilibrated conditions.

In the revised version of Manuscript and Supplementary Information, we have updated the results (Figure 5b-c and Figures S19-S21) and the description of the first-principles simulations (Pages S9 in Supplementary Information).

On page S9: “The simulation ran under a constant number of atoms, volume, and temperature (NVT) ensemble, as well as a constant number of atoms, pressure, and temperature (NPT) ensemble. Along the trajectory, we now initialize simulation with 3 ps NVT simulation for heating (near 0 K to 300-500 K), with 1 fs for each step and temperature controlled by a Nosé-Hoover thermostat,²⁴ then follow by 5 ps NPT to the target pressure and eventually run 10 ps NVT simulation for equilibrium. The standard deviations of pressure are generally less than 1 GPa. Reaching equilibrium generally takes 10^4 FPMD steps (10 ps), which is judged by the fluctuation of thermodynamical variables.”

On page 12: “The Lindemann melting parameters (δ) of Cu₂ atoms were 0.128, 0.206, and 0.245 at 0, 9.4, and 13.3 GPa, respectively.” and “Although both the temperature and pressure tend to stimulate vibration and motion of all atoms, the pressure more selectively regulates the local bonding to induce disordered Cu₂ sublattice that creates the NOF structure (Figures 5c and S21).”

On page 12:

Figure 5. (b) The time-dependent mean square displacement (MSD) at 0 GPa and 9.4 GPa at room temperature. The dash and shade lines are the average MSD and melting threshold MSD of Cu2 atoms, respectively. (c) The MSD of Cu2 atoms in $\text{Cu}_{12}\text{Sb}_4\text{S}_{13}$ at different pressures and temperatures.

On page S34:

Figure S19. The time-dependent MSD under 13.3 GPa at 300 K. The dash and shade lines are average MSD and melting threshold MSD of Cu2 atoms, respectively.

On page S35:

Figure S20. The time-dependent MSD at selected pressures at 400 K and 500 K. The dash and shade lines are average MSD and melting threshold MSD of Cu2 atoms, respectively.

On page S36:

Figure S21. The MSD of Cu1 and Sb atoms in $\text{Cu}_{12}\text{Sb}_4\text{S}_{13}$ at different pressures and temperatures.

10. In Table S1, the reported lattice parameters correspond to the experimental data or to the relaxed DFT structures? A comparison is needed.

Response: Thanks for pointing this out. The reported lattice parameters in Table S1 are experimental data. We added the relaxed DFT structures in revised Supplementary Information, which are consistent with the experimental data.

The details about the DFT relaxed structures have been added in the revised version of Supplementary Information.

On page S28:

Table S2. DFT relaxed crystal structures of $\text{Cu}_{12}\text{Sb}_4\text{S}_{13}$ at selected pressures.

Pressure (GPa)	a (Å)	V (Å ³)
0	10.38713	1120.694
1.9	10.22620	1069.406

3.3	10.16140	1049.205
5.1	10.07680	1023.217
7.2	9.97810	993.444
9.4	9.90270	971.093
11.3	9.84120	953.112
13.3	9.75300	927.715

11. In Figures S11 and S12, why the electronic structure at a pressure of 9.4 GPa is not presented? This value of the pressure seems to be critical from the rest of the results presented in the manuscript, hence it would have been significant to see the density of states (DOS) and investigate the behaviour of the electronic structure.

Response: Thanks for this comment. We put the electronic structure of 0, 5.1, and 7.2 GPa that want to explain the variation of electric conductivity under pressure. The effective masses of the electron decrease upon compression, which indicates enhanced carrier mobility. Such behavior can explain the increase of electrical conductivity up to 7 GPa.

According to the reviewer's suggestion, electronic structure and DOS of 9.4 GPa have been included in the revision as Figures S15 and S16, respectively. The effective masses of the electron decrease up to $0.64 m_0$ at 9.4 GPa, which indicates enhanced carrier mobility. However, the part-disordered structural transition occurs at above 9.4 GPa. Thus, electron scattering enhances which plays a significant role in inhibiting charge transport.

On page S12: "The effective masses of the electron decrease upon compression ($1.61 m_0 \rightarrow 0.64 m_0$ from 0 to 9.4 GPa), which indicates enhanced carrier mobility (Figure S15). Because of the rising carrier mobility, the electrical conductivity increases up to 7 GPa.

However, due to the part-disordered structural transition, electron scattering enhances and brings the decrease of electrical conductivity above 9.4 GPa.”

On page S29:

Figure S15. The calculated electronic structures of $\text{Cu}_{12}\text{Sb}_4\text{S}_{13}$ at 0, 5.1, 7.2, and 9.4 GPa.

On page S30:

Figure S16. The calculated density of states of $\text{Cu}_{12}\text{Sb}_4\text{S}_{13}$ at 0, 5.1, 7.2, and 9.4 GPa.

12. In Table S2, the changes of the Bader charges from 0 to 9.4 GPa pressures are insignificant. One should expect the changes in the atomic structure to be reflected on the Bader ionic charges (difference between crystal/glass).

Response: We would like to note that an obvious decrease in Bader charge from 0.40 at ambient pressure to 0.30 at 11.3 GPa of Cu₂ atoms can be observed (Table S3). This change is more significant in comparison to other atoms (Figure R9). Generally, owing to the more complex coordination and weaker covalence bonding, the amorphous state has a lower average Bader charge than the one in crystalline state.²⁴⁻²⁷ The previous studies reported a 0.1–0.3e difference between the crystalline and amorphous states.²⁴⁻²⁶ Thus, the changes in the Bader charge of Cu₂ are reflected in the difference between the NOF and crystalline structures. Furthermore, we used the Bader charge to calculate bond orders. The enhanced bond order between Cu₂ and Sb (from 0.10 at ambient pressure to 0.25 at 11.3 GPa) suggests the strong electrostatic repulsive force, which shoves Cu₂ atoms away from the equilibrium position and leads to the disordered Cu sublattice.

Figure R9. The normalized Bader charge shows the change degree of Bader charge in each atom under pressures.

The details about Bader charges and bond orders at 11.3 GPa have been added in the revised version of Supplementary Information.

On page S31:

Table S3. Bader charges and bond orders of different atoms and atom pairs at different pressures.

	Pressure (GPa)	Cu2	Cu1	Sb	average S
Bader charge	0	0.40	0.49	1.03	-0.75
	7.2	0.35	0.47	1.05	-0.72
	9.4	0.34	0.46	1.06	-0.70
	11.3	0.30	0.47	1.03	-0.71
	Pressure (GPa)	Cu2–Sb	Cu1–S	Cu2–S	Sb–S
Bond order	0	0.10	0.63	0.72	0.84
	7.2	0.20	0.70	0.78	0.86
	9.4	0.22	0.72	0.80	0.87
	11.3	0.25	0.75	0.84	0.83

13. In Figure S13, the three atomistic structures look, again, identical. I cannot see any difference. The figures do not provide any useful information.

Response: We thank the reviewer for pointing this out. In the revised version, we have zoomed in the trajectory of typical Cu2 atoms to clearly show the enhanced anharmonic motion of Cu2 atoms.

Figure S18 about MD trajectory has been modified in the revised Supplementary Information.

On page S33:

Figure S18. The trajectory of Cu atoms at selected pressures and temperatures.

References:

1. McMahon, M., Nemes R. Chain “melting” in the composite Rb-IV structure. *Phys. Rev. Lett.* **93**, 055501 (2004).
2. McBride, E. E., *et al.* One-dimensional chain melting in incommensurate potassium. *Phys. Rev. B* **91**, 144111 (2015).
3. Robinson, V. N., Zong H., Ackland G. J., Woolman G., Hermann A. On the chain-melted phase of matter. *Proc. Natl. Acad. Sci. U.S.A.* **116**, 10297-10302 (2019).
4. Chapman, K. W., Halder G. J., Chupas P. J. Guest-dependent high pressure phenomena in a Nanoporous metal– organic framework material. *J. Am. Chem. Soc.* **130**, 10524-10526 (2008).
5. Lee, Y., Kim S. J., Kao C.-C., Vogt T. Pressure-Induced Hydration and Order– Disorder Transition in a Synthetic Potassium Gallosilicate Zeolite with Gismondine Topology. *J. Am. Chem. Soc.* **130**, 2842-2850 (2008).
6. Fan, W., *et al.* A fluorine-functionalized microporous In-MOF with high physicochemical stability for light hydrocarbon storage and separation. *Inorg Chem Front* **5**, 2445-2449 (2018).
7. Shamblin, J., *et al.* Probing disorder in isometric pyrochlore and related complex oxides. *Nat. Mater.* **15**, 507-511 (2016).
8. Blanchard, P. E., *et al.* Does local disorder occur in the pyrochlore zirconates? *Inorg.*

Chem. **51**, 13237-13244 (2012).

9. Yashima, M., Sasaki S., Kakihana M., Yamaguchi Y., Arashi H., Yoshimura M. Oxygen-induced structural change of the tetragonal phase around the tetragonal–cubic phase boundary in ZrO_2 – $YO_{1.5}$ solid solutions. *Acta. Crystallogr. B. Struct. Sci. Cryst. Eng. Mater.* **50**, 663-672 (1994).

10. Jiang, Z., *et al.* Atomic interface effect of a single atom copper catalyst for enhanced oxygen reduction reactions. *Energy Environ. Sci.* **12**, 3508-3514 (2019).

11. Shang, H., *et al.* Engineering unsymmetrically coordinated Cu-S₁N₃ single atom sites with enhanced oxygen reduction activity. *Nat. Commun.* **11**, 1-11 (2020).

12. Prescher, C., Prakapenka V. B. DIOPTAS: a program for reduction of two-dimensional X-ray diffraction data and data exploration. *High Pressure Res* **35**, 223-230 (2015).

13. Kitagawa, S., *et al.* Suppression of nonmagnetic insulating state by application of pressure in mineral tetrahedrite $Cu_{12}Sb_4S_{13}$. *J. Phys. Soc. Japan* **84**, 093701 (2015).

14. Nasonova, D. I., Verchenko V. Y., Tsirlin A. A., Shevelkov A. V. Low-temperature structure and thermoelectric properties of pristine synthetic tetrahedrite $Cu_{12}Sb_4S_{13}$. *Chem. Mater.* **28**, 6621-6627 (2016).

15. Zhu, C., *et al.* Improved Thermoelectric Performance of $Cu_{12}Sb_4S_{13}$ through Gd-Substitution Induced Enhancement of Electronic Density of States and Phonon Scattering. *ACS Appl. Mater. Interfaces* **13**, 25092-25101 (2021).

16. Huang, L., *et al.* Effects of Sb Deviation from Its Stoichiometric Ratio on the Micro-and Electronic Structures and Thermoelectric Properties of $Cu_{12}Sb_4S_{13}$. *ACS Appl. Mater. Interfaces* **12**, 14145-14153 (2020).

17. Bolz, L., Mauer F. Measurement of the Lattice Constants of Neon Isotopes in the Temperature Range 4–24° K. *Advances in X-ray Analysis* **6**, 242-249 (1962).

18. Kolobov, A., Krbal M., Fons P., Tominaga J., Uruga T. Distortion-triggered loss of long-range order in solids with bonding energy hierarchy. *Nat. Chem.* **3**, 311-316 (2011).

19. Simpson, R. E., *et al.* Interfacial phase-change memory. *Nat. Nanotechnol.* **6**, 501-505 (2011).

20. Raoux, S., Xiong F., Wuttig M., Pop E. Phase change materials and phase change

memory. *MRS Bull.* **39**, 703-710 (2014).

21. Vopson, M. M., Rogers N., Hepburn I. The generalized Lindemann melting coefficient. *Solid State Commun* **318**, 113977 (2020).

22. Vinet, P., Ferrante J., Rose J. H., Smith J. R. Compressibility of solids. *J. Geophys. Res. Solid Earth* **92**, 9319-9325 (1987).

23. Lai, W., Wang Y., Morelli D. T., Lu X. From bonding asymmetry to anharmonic rattling in $\text{Cu}_{12}\text{Sb}_4\text{S}_{13}$ tetrahedrites: When lone-pair electrons are not so lonely. *Adv. Funct. Mater.* **25**, 3648-3657 (2015).

24. Wu, M., Tse J. S., Wang S., Wang C., Jiang J. Origin of pressure-induced crystallization of $\text{Ce}_{75}\text{Al}_{25}$ metallic glass. *Nat. Commun.* **6**, 1-7 (2015).

25. Hu, S., Liu B., Li Z., Zhou J., Sun Z. Identifying optimal dopants for Sb_2Te_3 phase-change material by high-throughput ab initio calculations with experiments. *Comput. Mater. Sci.* **165**, 51-58 (2019).

26. Mocanu, F. C., Konstantinou K., Mavračić J., Elliott S. On the Chemical Bonding of Amorphous Sb_2Te_3 . *Phys Status Solidi Rapid Res Lett* **15**, 2000485 (2021).

27. Caravati, S., Bernasconi M., Kühne T., Krack M., Parrinello M. First-principles study of crystalline and amorphous $\text{Ge}_2\text{Sb}_2\text{Te}_5$ and the effects of stoichiometric defects. *J. Phys. Condens. Matter* **21**, 255501 (2009).

REVIEWER COMMENTS

Reviewer #1 (Remarks to the Author):

The Authors have addressed all questions, which were raised by the Reviewers.

By inspecting .cif files provided by the Authors I noticed that only the ADPs of Cu2 were refined. Furthermore, there are no standard deviations of refined coordinates of Sb1 and S atoms. This leads to the question of how were these coordinates obtained and why ADPs of Cu1, Sb1, S1 and S2 left unrefined?

Until a proper Rietveld refinement is performed, the Figure S2 can not be used for the discussion in the manuscript.

Reviewer #2 (Remarks to the Author):

In the response the authors considered seriously all the comments made by the reviewers. They put some significant effort to accommodate all the requests and concerns raised by the reviewers and to provide sufficient explanations.

The revised Introduction provides more content regarding the research activities in the literature and the importance of the design of such hybrid ordered-disordered structures. Moreover, the authors present in a more lucid way their choice of applying pressure in this study as means of modifying the atomic structure.

I believe the new Fig. 3 is a strong addition in the manuscript. The authors performed two new experiments, using X-ray absorption spectroscopy and scanning-transmission electron microscopy, to investigate the atomic-scale structure of the system under study before and after the application of a high pressure. The experimental data and the respective analysis presented in this figure provide a strong indication about the formation of the NOF structure.

The revised schematic illustration presented in Fig. 1b seems to be more appropriate, while the behaviour described in the new drawing can also be reflected in the (new) experimental and simulation data.

The revised Fig. 2, which includes the ELF line profile (Fig. 2f) shows clearly the different behaviour under pressure and highlights the argument of the authors about the suppression of the lone-pair electrons. In that way, it is also more clear to the readers how the volume of the isosurface is reduced (i.e. weaker localization) after the application of the external high pressure (Fig. 2e).

In addition, the authors performed new atomistic (DFT) simulations. They included a constant pressure (NPT) molecular-dynamics run in their simulation protocol, while they also extended the collection-data trajectories from 5ps to 10ps. I appreciate their efforts and I believe the new DFT simulations have increased the quality of their study. It is important to highlight that the Mean Squared Displacement (MSD) is an averaged quantity, therefore the size of the modelled system (i.e. number of atoms) and the length of the molecular-dynamics trajectory (i.e. simulation time) do play a role in the calculated MSD (the comment about the total number of electrons the authors made in their response it is not really relevant to the MSD calculation). The presentation of the results in the new Fig. 5 (and in the relevant figures in the Supplementary Information) is more clear, while also I believe that the analyses from the extended trajectories highlight better the behaviour under pressure from the quantitative point of view (better statistics), and hence strengthens the arguments. Moreover, the authors presented a new calculation for the vibrational density of states relating to their discussion about the dynamical behaviour of the atoms in the structure before and after the application of high pressure.

Also, the authors extended their calculations, analyses and discussions about the electronic structure of the system for values of higher applied pressure, as well as the same for the Bader-charges analysis.

Finally, the authors provided some raw data as Supplementary material upon request of the Reviewer #1.

Overall, I was pleased to see that the new experiments and simulations performed by the authors, together with the revisions made in the original manuscript have enhanced the validity of the arguments presented in the study and increased its quality. Therefore, I would like to recommend the publication of this revised manuscript in Nature Communications.

However, I would like to suggest an alteration in the title of the paper. I still think that the term “amorphous” does not correspond to the right description for the formation of the distorted Cu₂ atoms inside the crystalline framework. Hence, I think the word should be replaced in the title accordingly. Instead of “amorphous innards”, it should be “distorted innards” or (maybe) “amorphous-like innards”. I think that even the authors, based on their answers in the response document, should agree with such change.

Point-by-point Responses to the Reviewers' Comments

(Manuscript ID: NCOMMS-21-44547A)

Reviewer #1:

The Authors have addressed all questions, which were raised by the Reviewers.

By inspecting .cif files provided by the Authors I noticed that only the ADPs of Cu2 were refined. Furthermore, there are no standard deviations of refined coordinates of Sb1 and S atoms. This leads to the question of how were these coordinates obtained and why ADPs of Cu1, Sb1, S1 and S2 left unrefined?

Until a proper Rietveld refinement is performed, the Figure S2 cannot be used for the discussion in the manuscript.

Response: We thank the reviewer very much for recognizing our efforts and improvement made on the revised manuscript. For the remaining concern regarding the refinement, we would like to note that in the previous refinements, we first refined all the atoms coordinates, lattice parameters, FWHM parameters and so on; then we refined the coordinates and ADPs of Cu2 atoms only with other atoms fixed to ensure the convergence of Rietveld refinement process. That is why there were no standard deviations of refined coordinates of Sb1 and S atoms and were no refined ADP of Cu1, Sb, and S atoms in the previous cif files.

According to reviewer's comments, we agree that refining the coordinates and ADPs of all atoms are more appropriate. In the revised cif files, we have refined the coordinates and ADPs of all the atoms (Cu2, Cu1, Sb, S1, and S2) step by step. Note that the refinements of crystal structures are instable beyond 9.4 GPa, where some ADPs have negative values. Thus, we refined the coordinates and ADPs of all atoms up to 9.4 GPa; beyond 9.4 GPa, we only refined the coordinates. Figure S2 has been updated based on the new refinement results, where the pressure-induced variations of ADPs are in consistent.

In the revised manuscript, we have updated all the XRD results based on the new refinements (Figures 2c, 2d, S1 and S2, table S1), which are consistent with the previous results.

Figure S2. Cu2 atomic displacement parameters (ADP) of $\text{Cu}_{12}\text{Sb}_4\text{S}_{13}$ at different pressures. The ADP values are derived from the Rietveld refinement of synchrotron X-ray diffraction. The ADP values of Cu2 increase from 0.020 to 0.064 \AA^2 during compression.

Reviewer #2:

In the response the authors considered seriously all the comments made by the reviewers. They put some significant effort to accommodate all the requests and concerns raised by the reviewers and to provide sufficient explanations.

The revised Introduction provides more content regarding the research activities in the literature and the importance of the design of such hybrid ordered-disordered structures. Moreover, the authors present in a more lucid way their choice of applying pressure in this study as means of modifying the atomic structure.

I believe the new Fig. 3 is a strong addition in the manuscript. The authors performed two new experiments, using X-ray absorption spectroscopy and scanning-transmission electron microscopy, to investigate the atomic-scale structure of the system under study before and after the application of a high pressure. The experimental data and the respective analysis presented in this figure provide a strong indication about the formation of the NOF structure.

The revised schematic illustration presented in Fig. 1b seems to be more appropriate, while the behaviour described in the new drawing can also be reflected in the (new) experimental and simulation data.

The revised Fig. 2, which includes the ELF line profile (Fig. 2f) shows clearly the different behaviour under pressure and highlights the argument of the authors about the suppression of the lone-pair electrons. In that way, it is also more clear to the readers how the volume of the isosurface is reduced (i.e. weaker localization) after the application of the external high pressure (Fig. 2e).

In addition, the authors performed new atomistic (DFT) simulations. They included a constant pressure (NPT) molecular-dynamics run in their simulation protocol, while they also extended the collection-data trajectories from 5ps to 10ps. I appreciate their efforts and I believe the new DFT simulations have increased the quality of their study. It is important to highlight that the Mean Squared Displacement (MSD) is an averaged quantity, therefore the size of the modelled system (i.e. number of atoms) and the length of the molecular-dynamics trajectory (i.e. simulation time) do play a role in the calculated MSD (the comment about the total number of electrons the authors made in their response it is not really relevant to the MSD calculation). The presentation of the results in the new Fig. 5 (and in the relevant figures in the Supplementary Information) is more clear, while also I believe that the analyses from the extended trajectories highlight better the behaviour under pressure from the quantitative point of view (better statistics), and hence strengthens the arguments. Moreover, the authors presented a new calculation for the vibrational density of states relating to their discussion about the dynamical behaviour of the atoms in the structure before

and after the application of high pressure.

Also, the authors extended their calculations, analyses and discussions about the electronic structure of the system for values of higher applied pressure, as well as the same for the Bader-charges analysis.

Finally, the authors provided some raw data as Supplementary material upon request of the Reviewer #1.

Overall, I was pleased to see that the new experiments and simulations performed by the authors, together with the revisions made in the original manuscript have enhanced the validity of the arguments presented in the study and increased its quality. Therefore, I would like to recommend the publication of this revised manuscript in Nature Communications.

Response: We appreciate the reviewer's positive comments and the recommendation of our manuscript for publication in Nature Communications. We are also very grateful for the reviewer's recognition of our efforts (new experiments, modified simulations, and extended discussion) made on the improvement of the manuscript.

However, I would like to suggest an alteration in the title of the paper. I still think that the term "amorphous" does not correspond to the right description for the formation of the distorted Cu₂ atoms inside the crystalline framework. Hence, I think the word should be replaced in the title accordingly. Instead of "amorphous innards", it should be "distorted innards" or (maybe) "amorphous-like innards". I think that even the authors, based on their answers in the response document, should agree with such change.

Response: We thank the reviewer very much for such an insightful suggestion. We agree with the reviewer that the "amorphous-like innards" is a more accurate description and we have revised "amorphous innards" into "amorphous-like innards" in both the title and main text of the revised manuscript.

REVIEWERS' COMMENTS

Reviewer #1 (Remarks to the Author):

The revised version of the manuscript addresses all concerns raised by the Reviewers. I recommend the publication in Nature Communications.